# Microgliopathy as a primary mediator of neuronal death in models of Friedreich's Ataxia

Carla Pernaci[1,2], Avalon Johnson[1,2], Sydney Gillette[1,2], Anna S. Warden[1,2], Chad McCormick[1,2], Sammy Weiser-Novak[3], Gabriela Ramirez[1,2], Emily H. Broersma [1,2], Priyanka Mishra[4], Anusha Sivakumar [4], Stephanie Cherqui[4,5] & Nicole G. Coufal [1,2,5] ✉

Friedreich's ataxia (FRDA) is an incurable neurodegenerative disorder caused by a GAA repeat expansion in the frataxin (*FXN*) gene, leading to a severe reduction of the mitochondrial FXN protein, crucial for iron metabolism. While microglial inflammation is observed in FRDA, it remains unclear whether immune dysfunction is a primary disease mediator or a secondary reactionary phenotype. Utilizing patient-derived induced pluripotent stem cells (iPSCs), we report an intrinsic microglial phenotype of stark mitochondrial defects, iron overload, lipid peroxidation, and lysosomal abnormalities. These factors drive a pro-inflammatory state that contributes to neuronal death in co-culture systems. In a murine xenograft model, transplanted human FRDA microglia accumulate in white matter and the Purkinje cell layer, resulting in Purkinje neuron loss in otherwise healthy brains. Notably, CRISPR/Cas9-mediated correction of the GAA repeat reverses microglial defects and mitigates neurodegeneration. Here, we suggest that microglial dysfunction serve as a disease driver and a promising therapeutic target in FRDA.

Friedreich's ataxia (FRDA) is a progressive neurodegenerative disease that affects 1 in every 50,000 individuals in the United States[1]. This autosomal recessive disorder is caused by a homozygous GAA trinucleotide expansion in the first intron of the *frataxin* (*FXN*) gene, impairing transcription initiation, and resulting in a drastic reduction in FXN protein production. FXN is a mitochondrial protein involved in iron-sulfur (Fe-S) cluster biosynthesis and energy production[2,3]. Reduced FXN expression results in a broad mitochondrial phenotype which encompasses reduced oxidative phosphorylation (OXPHOS), attenuated ATP production[2,3], reactive oxygen species (ROS) accumulation and iron overload[4–6]. Chronically this perpetuates a cycle of DNA damage and mitochondrial dysfunction in FRDA, exerting the strongest pathological phenotype in tissues that rely on higher energy consumption, especially the nervous system. No cure is currently available to treat this devastating progressive neurodegenerative condition.

The cell-type-specific contribution to the pathophysiology of neurodegeneration in FRDA is incompletely understood, hampering rational therapeutic targeting. Neurons are known to be susceptible to FXN loss, with neuronal degeneration being reported in post-mortem human samples and different FRDA mouse models[7–11]. In FRDA post-mortem tissue, degeneration of nerve fibers in the spinal cord leading to neuronal loss, degeneration of large neurons of the dorsal root ganglia, degeneration of dentate nucleus neurons and reduction in cerebellar peduncles have been reported, as well as patchy loss of Purkinje cells and inferior olivary nucleus neuronal degeneration[12]. We recently characterized the pathological phenotype of induced pluripotent stem cell (iPSC)-derived FRDA neurons and showed increased

[1]Department of Pediatrics, University of California San Diego, La jolla, CA, USA. [2]Sanford Consortium for Regenerative Medicine, La Jolla, CA, USA. [3]Salk Institute for Biological Studies, La Jolla, CA, USA. [4]Department of Pediatrics, Division of Genetics, University of California San Diego, San Diego, CA, USA. [5]These authors contributed equally: Stephanie Cherqui, Nicole G. Coufal. ✉e-mail: ncoufal@health.ucsd.edu

caspase-3 expression and a higher incidence of dendritic blebbing, both distinct indications of neuronal apoptosis[8]. Additionally, FRDA neurons demonstrated higher levels of mitochondrial superoxides, abnormal mitochondrial ultrastructure and aberrant $Ca^{2+}$ signaling. Our dual guide CRISPR/Cas9 gene editing successfully restored *FXN* transcript levels in gene-edited neurons and promoted neuroprotection[8]. This genome editing strategy was previously optimized to remove the hyperexpansion mutation in human $CD34^+$ hematopoietic stem and progenitor cells isolated from FRDA, aiming to advance this approach toward a clinical trial[13].

Beyond neurons, there is increasing evidence of glial involvement in FRDA, yet whether this is primary or a secondary response to neuronal death is unknown. In physiological conditions, microglia, the resident macrophages of the central nervous system (CNS), play both an immune and neuroprotective role as they provide constant surveillance of the environment and contribute to neurogenesis and synapse homeostasis[14,15]. Diverse CNS insults cause microglial overactivation, negatively impacting neurons via the release of neurotoxic factors or dysfunctional phagocytosis[16], ultimately contributing to chronic inflammation and neurodegeneration. FRDA patients demonstrate an increase in ferritin-positive hypertrophic microglia, with thicker processes in the dentate nucleus. Elevated microglia activation in dentate nuclei, brainstem, and cerebellar cortex of FRDA patients is correlated with earlier disease onset, shorter disease duration, and higher levels of inflammatory cytokines such as plasma IL6; highlighting chronic neuroinflammation as a potential key pathogenic factor in FRDA[17]. In the KIKO murine model of FRDA, cerebellar microglia show gene expression changes in pathways that regulate inflammation and oxidative stress, together with a metabolic shift from OXPHOS to elevated glycolysis, increased phagocytosis, impaired migration and overall disrupted microglia homeostasis[18]. Additionally, cerebellar microglia exhibit an imbalance in immunometabolism, characterized by a loss of homeostasis, hyperactivation, and increased glucose metabolism[19]. Similarly, in the humanized YG8-800 FRDA murine model, early cerebellar glial activation has been reported with an increase in mRNA of key proinflammatory cytokines, leading to cerebellar atrophy with loss of Purkinje cells[20]. However, none of these studies have queried microglial dysfunction in the absence of neuronal damage, such that it is currently unknown whether microglial reactivity is a primary mediator of neurodegeneration or a secondary reaction to progressive neuronal dysfunction and neuronal death.

Here, to untangle the impact of FXN deficiency specifically in microglia and to disclose microglial contribution to disease pathogenesis in FRDA, we used iPSC-derived microglia (iMGs) from FRDA patients, related familial carriers and unrelated healthy donors and applied the previously reported CRISPR/Cas9 gene editing approach to explore its therapeutical potential in our experimental conditions. In this work, we show that FRDA microglia exhibit a strong cell-autonomous phenotype in vitro, including a hyperinflammatory activation profile, stark mitochondrial and lysosomal defects, and marked imbalance of iron, lipid and ROS levels, eventually terminating in marked DNA damage. In a 2D co-culture model, we demonstrate that the addition of FRDA microglia to healthy neurons resembles some patient-derived FRDA neuron phenotypes, including neuronal bleb formation and caspase 3 activation. We then extended our findings in vivo, where xenotransplantation of microglia progenitors into a unique humanized murine model depleted of endogenous microglia revealed a hyperinflammatory microglia phenotype in the cerebellum of adult mice. FRDA microglia accumulate within the Purkinje cell layer and lead to ROS-mediated DNA damage in healthy Purkinje cells and cerebellar neuronal death in vivo. All the above alterations were strongly attenuated by the GAA gene editing therapeutic approach in microglia. Altogether, these findings identify a microglia role in the pathogenesis of neurodegeneration in FRDA and suggest the therapeutic potential for a dual-guide gene editing strategy.

## Results

### FXN deficiency leads to an activated microglial phenotype in vitro

We first queried whether there is evidence of microglial activation in FRDA patients, comparing postmortem cerebellum to that of healthy controls (Fig. 1a), finding that $Iba1^+$ microglia in the cerebellar molecular layer of FRDA patients exhibit decreased ramification complexity consistent with a more amoeboid morphology (Fig. 1b). We next examined an existing transcriptomic dataset[21] of healthy primary human fetal and postnatal microglia to identify whether FXN is expressed in healthy human microglia, finding that the FXN transcript in microglia is equivalent to that in whole cortex (Supplementary Fig. 1a).

Since microglia-specific *FXN* deletions have not been feasible in murine models, it has been difficult to investigate the microglial-specific contribution to FRDA pathology. We therefore queried the potential validity of applying iPSC models to FRDA, again utilizing a published transcriptomic dataset[21], finding that FXN expression is retained across several iPSC-derived models, including human iMG and xenotransplanted microglia (xMG) (Supplementary Fig. 1a). We next applied an existing cohort of FRDA patient, unrelated control, and dual guide CRISPR/Cas9 *FXN* repeat gene edited iPSCs[8], and generated additional asymptomatic familial carrier cell lines heterozygous for the GAA *FXN* repeat expansion as immediate familial controls of FRDA patients within this cohort (Fig. 1c and Supplementary Fig. 1b-e). Utilizing this unique iPSC cohort, we sought to understand whether human iMGs from FRDA patients exhibit a cell-autonomous disease-specific phenotype. We differentiated iMGs (Fig. 1c, Supplementary Figs. 1f, and 2a–c) from healthy donors, FRDA patients, familial heterozygous carriers (henceforth referred as carriers) and *FXN* CRISPR/Cas9 gene edited lines (henceforth referred to as edited) to explore the therapeutic potential of GAA trinucleotide repeat expansion correction specifically in iMGs. We did not identify differences in the differentiation efficacy or basal microglial/myeloid marker expression between lines, including in CX3CR1, CD11b and CD45 (Supplementary Fig. 1g, h). Additionally, microglia from all cell lines expressed comparable levels of the mature type-I transmembrane microglia marker TMEM119 (Supplementary Fig. 2a)

Given successful microglial differentiation, we next examined FXN protein levels, finding decreased FXN in FRDA patients, and confirmed the efficacy of our gene editing strategy which restored FXN protein levels in gene-edited iMGs (Fig. 1d). Furthermore, co-labeling of mature iMGs with FXN and Tom20, a mitochondrial marker, revealed appropriate repopulation of $Tom20^+$ mitochondria with FXN in edited lines compared to their respective isogenic FRDA lines (Fig. 1e). Exploring the putative cell-autonomous microglia inflammatory phenotype in vitro, we leveraged a high throughput automated image analysis approach, finding that FRDA iMGs exhibit morphological differences with reduced ramification length and branch points suggestive of an activated profile[22] compared to healthy controls (Fig. 1f, Supplementary Fig. 2b). Additionally, we used CD68 and Iba1 levels as proxy for elevated microglia inflammatory states[23,24]. Previous studies have shown an increase in both CD68 and Iba1 after inflammation[25,26] and indeed we noted an increase in both markers in response to LPS stimulation[1] (Supplementary Fig. 2d). Analysis of FRDA microglia identified elevated levels of CD68 (Fig. 1g, top panel and Supplementary Fig. 2c), a lysosomal protein associated with microglial activation and elevated phagocytic capacity[27] as well as increased Iba1 intensity (Fig. 1g, bottom panel Supplementary Fig. 2c). Importantly, FXN protein restoration in gene-edited lines attenuated the FRDA microglia inflammatory state in vitro (Fig. 1f, g).

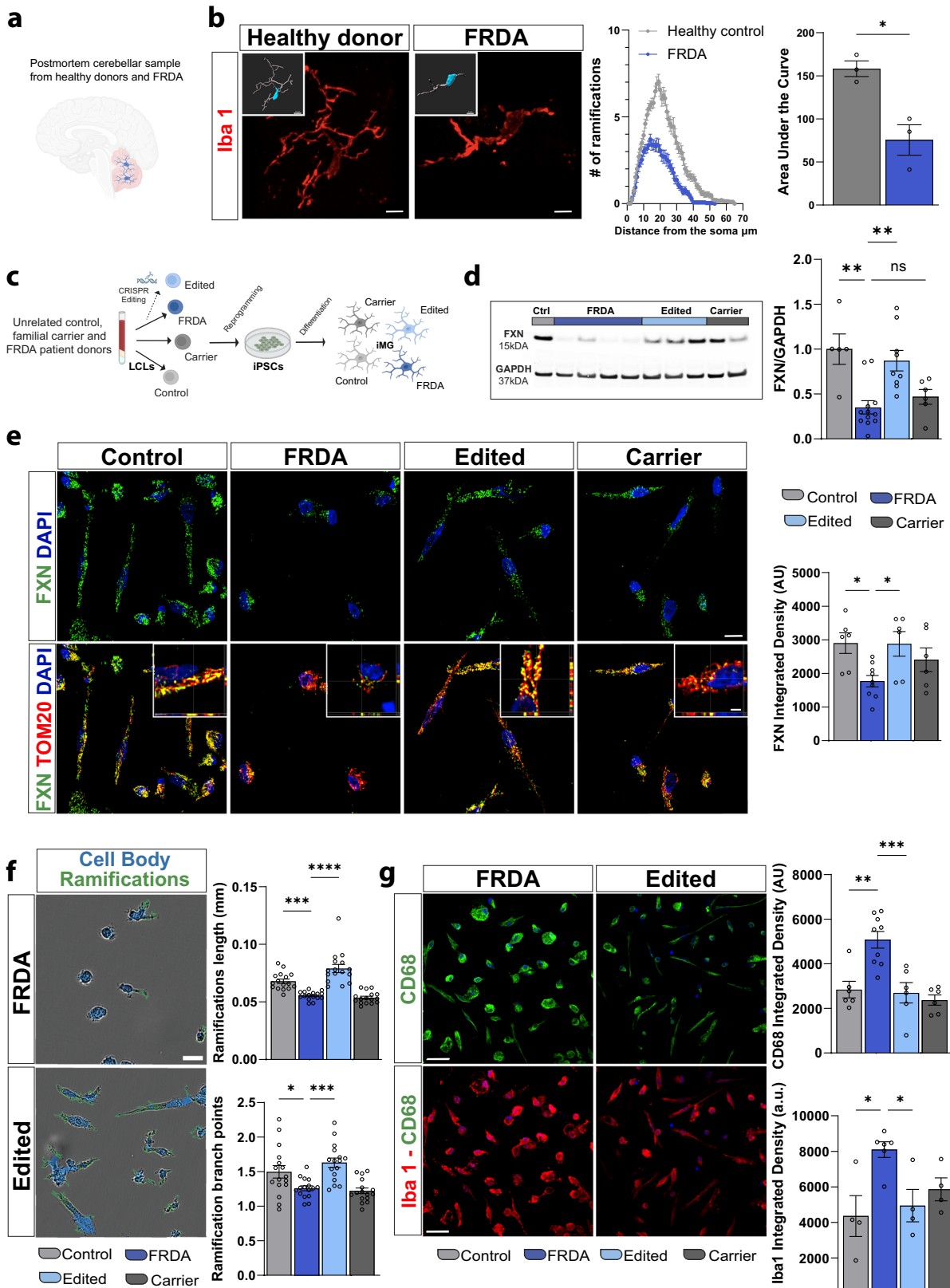

## Mitochondrial deficits in FRDA iMG are restored by gene editing

A decrease in FXN expression has been associated with increased mitochondrial oxidative stress and impaired ETC function in various cellular models of FRDA pathology[3,7,28]. Mitochondria are a significant source of ROS production within the cell, where an activated state can lead to aberrant ROS accumulation, exacerbating oxidative stress and cellular damage[7]. We quantified mitochondrial-specific ROS (MitoSox), observing accumulation within the mitochondria and a pronounced enrichment of ROS in FRDA iMG (Fig. 2a, b and Supplementary Fig. 3a), a finding consistent with our recent report in FRDA neurons[8]. Accumulation of mitochondrial ROS has detrimental consequences on mitochondrial biogenesis and ATP production by perturbation of

**Fig. 1 | FRDA microglia exhibit an activated phenotype and morphology.**
**a** Postmortem tissue from FRDA patients and healthy donors, $N = 3$/group.
**b** Microglial immunofluorescence in post-mortem cerebellar sections from three
FRDA patients and three healthy donors (left). Sholl analysis of microglia ramifi-
cations and soma size (middle) and area under the curve (AUC) quantification
(right). **c** Schematic overview for generation of an iPSC derived microglia (iMG)
cohort created in *BioRender. Coufal, N.* (2025) https://BioRender.com/ahj5pyk
**d** Western blot (left) and quantitation (right) depicting FXN protein expression in
iMGs normalized to GAPDH. **e** Immunocytochemistry (left) and quantification
(right) of FXN expression in iMG. **f** Representative images (left) and quantitation
(right) of high-content image analysis of iMG ramification length and branch points.

**g** Immunocytochemistry (left) and quantitation (right) of microglial lysosomal
protein CD68 and Iba1. Each dot in the plot represents a biological replicate with
$N = 3$ biologically independent replicates/differentiations for each iMG cell line.
Two controls, three FRDA, two gene-edited and two familial carrier lines were
utilized in (**e**, **g**) while in (**d**, **f**), two controls, four FRDA, three gene-edited and two
familial carrier lines. Integrated Density (AU: arbitrary units) has been quantified
per each cell, with at least 80 iMG cells/line/experiment analyzed. Data are pre-
sented as mean ± SEM, analyzed by one-way ANOVA with Tukey's post-hoc testing.
*$p < 0.05$, **$p < 0.01$, ***$p < 0.001$, ****$p < 0.0001$. Scale bars (**b**) 10 μm for healthy
donor and 8 μm for FRDA. Scale bars (**e**) 10 μm and 2 μm for zoomed-in section
view of mitochondrial FXN localization. Scale bars **f** 20 μm, **g** 30 μm.

mitochondrial membrane potential[29]. To clarify whether impairments
in mitochondrial membrane potential occur in FRDA microglia, we
utilized a mitochondrial potential-dependent dye (MitoTracker
DeepRed (MTDR). The results revealed significantly reduced MTDR in
FRDA microglia, thus suggesting compromised mitochondrial mem-
brane potential (Fig. 2c, d and Supplementary Fig. 3b). Elevated
mitochondrial oxidative stress may lead to mitochondrial deficits in
OXPHOS and mitochondrial function[2,5,30]. We therefore queried the
mitochondrial respiration capability of FRDA microglia. We measured
oxygen consumption with the Agilent Seahorse XF Mito Stress Test kit
and found that ATP production and mitochondrial respiratory capa-
city, both basal and maximal, are substantially reduced in FRDA
microglia (Fig. 2e, f, Supplementary Fig. 3c).

Given that reduced ATP production may lead to mitochondrial
fission[31,32], we hypothesized that functional mitochondrial impair-
ments coincide with mitochondrial structural abnormalities. To query
mitochondrial ultrastructure, we assessed mitochondrial morphology
and number using electron microscopy. We first found that in contrast
to our gene-corrected and healthy donor controls, microglia derived
from FRDA patients exhibited a higher overall number of mitochondria
as well as a higher proportion of spherical mitochondria, suggestive of
excessive mitochondrial scission (Fig. 2g, h). Furthermore, FRDA
mitochondria appeared to exhibit disrupted cristae structure (Fig. 2i).
Quantification of electron transport chain (ETC) complexes did not
identify significant differences in complex levels, unlike what has been
reported in other FRDA cell types[8,33] (Supplementary Fig. 3d), although
complex I and II trended lower, as has been previously reported[8,28].
Additionally, we did not find any significant differences in TOM20
protein levels (Supplementary Fig. 3e). Due to elevated ROS levels in
FRDA iMGs, we queried whether inducible nitric oxide levels also
accumulate in FRDA microglia, but did not find any significant differ-
ences between genotypes (Supplementary Fig. 3f). Collectively, our
data suggest that FXN loss in microglia has detrimental consequences
on mitochondrial morphology and biogenesis, a phenotype described
in other FRDA cellular models. Importantly, GAA repeat correction
strongly ameliorated the mitochondrial phenotype that characterizes
FRDA microglia in all assays.

## Dysregulated mitochondria-lysosome interplay in FRDA iMG

Activity-dependent inter-organelle signaling pathways facilitate the
recycling of impaired cellular components to sustain overall cellular
homeostasis[34]. Mitochondria-lysosomal interplay regulates a wide
array of cellular functions and ensures the effective elimination of
nonfunctional mitochondria through mitophagy; a lysosomal-
dependent type of autophagy. Under physiological conditions, mito-
phagy is crucial for sustaining microglial homeostasis by regulating
their inflammatory states and activity[35,36]. We thus tested the efficacy of
the mitophagy machinery using the mitophagy dye MD1-10, which
fluoresces at pH4-5 when the mitochondria (red) fuse with the acidic
lysosomes (green). Due to a lack of signal in all lines without a stimulus,
we investigated mitochondrial stress through treatment with carbonyl
cyanide m-chlorophenylhydrazone (CCCP) 50 mM. Three hours after
treatment, we found that FRDA iMGs have a reduced capacity to

induce mitophagy when compared to controls, as reflected by a
reduction in yellow colocalization, indicating the correct phagosome-
lysosome fusion occurring during mitophagy (Fig. 3a and Supple-
mentary Fig. 4a). This finding suggests that the accumulation of round
and dysfunctional mitochondria in FRDA iMGs may partly depend on
altered lysosomal-dependent type of autophagy.

Proper mitophagy in microglia alleviates microglial inflammatory
responses and its alterations may result in hyperinflammation and
perturbed microglia activity[35,36]. Given the hyperinflammatory micro-
glia morphology and marker expression, we next interrogated micro-
glia activity by examining the phagocytic capacity of iMGs, finding that
FRDA iMGs are hyperphagocytic of Zymosan particles (Fig. 3b, c and
Supplementary Fig. 4b). Moreover, the microglial TAM receptor tyr-
osine kinase, AXL, which is upregulated in phagocytic microglia[37], was
strongly enriched in FRDA iMGs compared to healthy controls
(Fig. 3d). Since FRDA microglia exhibited attenuated lysosomal
dependent mitophagy, we queried whether mitochondria impair-
ments might coincide with lysosomal defects, as reported in other
types of neurodegenerative disorders[38,39]. To this end, we first queried
lysosomal-associated membrane protein 1 (LAMP1) and found an
overall increase in total and glycosylated levels of LAMP1 in FRDA lines
compared to healthy controls (Supplementary Fig 4c). Interestingly,
microglial expression of glycosylated LAMP1 has been correlated with
Purkinje cell neurodegeneration[40]. Further, staining of iMGs with the
lysosome-specific dye, LysoTracker, showed that FRDA iMGs have
larger or a greater quantity of lysosomes (Fig. 3e and Supplementary
Fig. 4d). Electron microscopy also suggested an increased lysosomal
density in FRDA iMGs (Fig. 3f, data not quantified). Quantitative ana-
lysis of lysosomal pH using a pH-dependent dye (LysoSensor) found
that FRDA lysosomes exhibit deficits in lysosomal acidification (Fig. 3g
and Supplementary Fig. 4e). Finally, we employed a lysosomal activity
dye, confirming attenuated lysosomal enzymatic activity in FRDA iMGs
(Fig. 4h and Supplementary Fig. 4f). Collectively, these data demon-
strate that FRDA iMGs exhibit impaired mitophagy, increased phago-
cytic activity, and an abundance of dysfunctional lysosomes. Notably,
the deficit in FRDA iMGs was strongly ameliorated by our CRISPR/Cas9
gene editing approach in all assays.

## Organelle dyshomeostasis leads to a pathogenic microglial state

Iron overload is commonly reported in FRDA pathology due to the
pivotal role of the FXN protein in modulating iron concentrations and
in facilitating the biosynthesis of iron-sulfur (Fe-S) clusters within
mitochondria[41-44]. Proper handling of iron molecules within mito-
chondria is essential for the optimal functioning of respiratory chain
complexes, making this organelle the primary site of iron utilization in
cells. Lysosomes, in turn, regulate the trafficking and intracellular
distribution of iron; therefore, dynamic organelle communication
ensures proper iron-dependent metabolic pathways. Given the
reported iron accumulation in various FRDA models[34,44-46], we
queried iron levels in FRDA iMGs (FerroOrange) and found a significant
increase in iron accumulation, which was significantly ameliorated by
gene editing (Fig. 4a and Supplementary Fig. 5a). Excessive iron levels
can result in elevated production of ROS, leading to detrimental

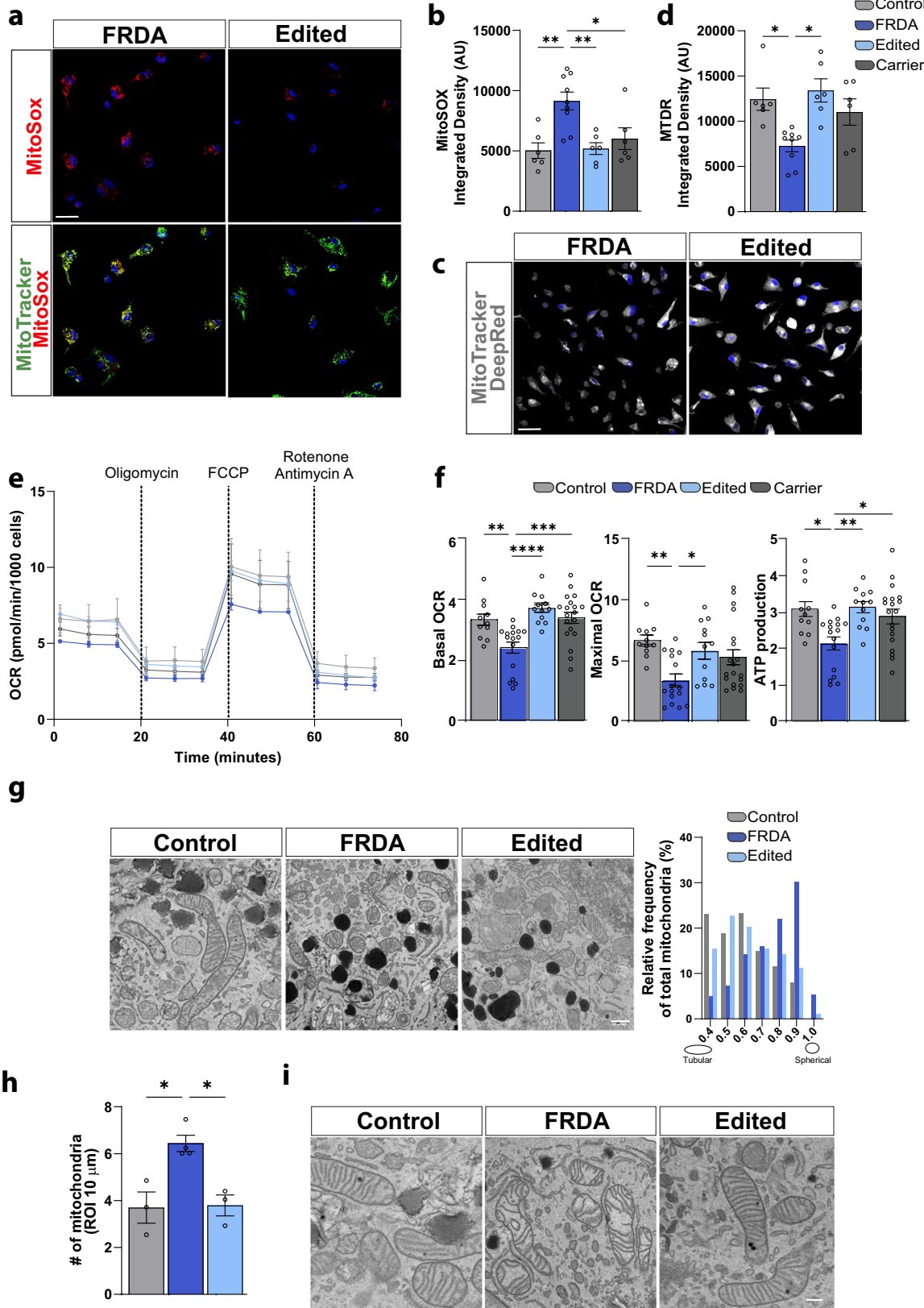

consequences on microglia-neuronal interactions. While we previously reported elevated mitochondrial ROS levels in our FRDA model, we also observed exacerbated overall cellular ROS in FRDA iMG compared to healthy controls (Fig. 4b and Supplementary Fig. 5b). The interplay between iron overload and excessive ROS production may disrupt lipid metabolism, negatively impacting microglial inflammatory responses[47]. Employing a neutral lipid stain (BODIPY493/503), we identified a marked accumulation of neutral lipids in FRDA iMGs, suggestive of improper lipid handling in diseased lines (Fig. 4c and Supplementary Fig. 5c). We next queried lipid peroxidation, a prominent feature of ferroptosis generating various oxidized byproducts that ultimately contribute to DNA damage. By using a fluorometric

**Fig. 2 | Mitochondrial dysfunction in FRDA microglia is corrected upon GAA gene editing. a** Representative confocal images of mitochondrial-specific reactive oxygen species (MitoSox) and mitochondria (MitoTracker) in FRDA and gene-edited microglia. **b** Quantitation of MitoSox accumulation in iMGs. **c** Assessment of mitochondrial membrane potential (MitoTrackerDeepRed (MTDR) in FRDA and gene-edited microglia. **d** Quantitation of MTDR by live cell imaging.
**e** Mitochondrial respiration in FRDA and gene-edited iMGs measured with the Seahorse Mito Stress Test, measuring oxygen consumption rate (OCR) in response to sequential injections of oligomycin, carbonyl cyanide 4-(trifluoromethoxy)phenylhydrazone (FCCP) and a combination of rotenone and antimycin A (left).
**f** Quantitation of mitochondrial respiration (right). **g** Electron microscopy (EM) images used for mitochondria organelle structural characterization. Sphericity analysis of individual mitochondria plotted as frequency distribution histogram (right). **h** Quantitation of mitochondria number in randomly assigned regions of interest (ROIs) of 10 μm². **i** Representative EM images of mitochondria cristae structure. Each dot in the plot represents a biological replicate with $N = 2–3$ independent experiments/line. Two controls, three FRDA, two gene-edited and two familial carrier lines/experiment (**a–f**). Integrated Density (AU: arbitrary units) has been quantified per each cell, with at least 80 iMG cells/line/experiment analyzed. For (**g, h**), three controls, three FRDA and two gene-edited lines are included and plotted in the analysis. Mitochondrial circularity was assessed on an average of 140 mitochondria/line, and for mitochondrial density, 153 different ROIs/line were included in the analysis. Data are presented as Mean ± SEM, and groups analyzed by one-way ANOVA with Tukey's post-hoc testing. *$p < 0.05$, **$p < 0.01$, ***$p < 0.001$, ****$p < 0.0001$. Scale bars **a** 30 μm, **c** 30 μm, **g** 40 pixels, **i** 50 pixels.

approach, where the switch of the fluorescent emission peak from red (neutral lipids) to green (peroxidized lipids) indicates oxidation of BODIPY 581/591 C11, we showed enriched peroxidized lipids in FRDA microglia as reflected by a reduced 591/581 ratio resulted by the pronounced increase in the green signal in the diseased microglia (Fig. 4d and Supplementary Fig. 5d). Oxidative stress and lipid peroxidation eventually lead to DNA damage[48]. We finally measured γH2AX as a proxy for ongoing DNA damage[49] and demonstrated abnormal γH2AX puncta accumulation in FRDA microglia (Fig. 4e). Collectively, this data suggests that dysregulation of organelle homeostasis has detrimental effects on iron and lipid metabolism in FRDA microglia, ultimately leading to excessive DNA damage. Notably, our gene editing approach effectively preserved microglial homeostasis and mitigated DNA damage throughout.

## FRDA microglia induce neurodegeneration of healthy neurons

The interplay of mitochondrial dysfunction, iron accumulation, and DNA damage in microglia results in a cascade of harmful effects, including oxidative stress, chronic inflammation, and neuronal injury, which may disrupt neuronal homeostasis, ultimately leading to neuronal dysfunction and degeneration[50]. To investigate how FRDA microglial dysfunction impacts neuronal viability, we developed an in vitro co-culture model by differentiating healthy neuronal progenitor cells (NPCs) into neurons and astrocytes, which were then co-cultured with varying genotypes of microglia at a 1:5 ratio (Fig. 5a, b). The co-culture composition was confirmed by staining for neurons (TUJ1: β-tubulin III), astrocytes (GFAP: glial fibrillary acidic protein) and microglia (Iba1) (Supplementary Fig. 6a).

Hyperinflammatory microglia have been demonstrated to promote neuronal iron uptake, leading to iron accumulation and ultimately cell death[51,52]. We utilized the live stain FerroOrange to measure intracellular iron accumulation in healthy neurons as a consequence of microglia co-culture. We found that the addition of FRDA microglia to the culture led to a significant increase in iron accumulation within healthy neurons (Fig. 5c). To further confirm that iron accumulation is specific to neurons and not astrocytes, we used a second neuronal model composed of iPSC derived induced neurons (iNs) generated through direct overexpression of the transcription factor NGN2 (Supplementary Fig. 6b). The addition of FRDA iMGs to healthy iNs resulted in a significant increase in iron accumulation within iNs (Supplementary Fig. 6c). An inflammatory environment has also been demonstrated to result in abnormal lipid accumulation in neurons[53], however no significant differences in neutral lipid accumulation (BODIPY493/503) were identified in co-cultured neurons (Supplementary Fig. 4b). To assess the consequences of elevated neuronal iron accumulation and overall microglial dysfunction, γH2AX foci were analyzed as a measure of double stranded DNA breaks. Healthy neurons co-cultured with FRDA iMGs demonstrated an increase in γH2AX puncta (Fig. 5d, e). Since FRDA microglia demonstrate increased ROS, the impact of oxidative stress was assessed in the healthy neuronal culture using 8-Hydroxydeoxyguanosine (8-OHdG), a marker of

oxidative DNA damage. A significantly higher number of healthy TUJ1+ neurons cultured with FRDA iMGs were positive for 8-OHdG (Fig. 5f, g). To better understand the consequences of this DNA damage, healthy co-cultured neurons were stained for cleaved caspase-3, a key executioner of apoptosis. An increase in caspase-3+ neurons was observed when co-cultured with FRDA microglia (Fig. 5h, i). An additional hallmark of caspase-3-mediated apoptosis is the formation of blebs or bulges of the neuronal cell membrane induced by the cleavage of ROCK-I by caspase3[54,55]. Increased neuronal blebbing was observed in neurons co-cultured with FRDA iMGs, reaffirming the occurrence of caspase-3-mediated neuronal apoptosis (Fig. 5j-k).

To further investigate the mechanism of this iMG-mediated neurodegeneration, microglia-conditioned media were collected and added to healthy neurons. γH2AX was measured in healthy TUJ1+ neurons to assess DNA damage induced through conditioned media treatment, finding no significant differences between conditions (Supplementary Fig. 6f, g). Caspase-3+ neurons were also quantified after treatment with conditioned media, and no significant differences were observed between experimental conditions (Supplementary Fig. 6h). Lastly, neuronal blebbing in healthy neurons was assessed also with no significant differences between conditions (Supplementary Fig. 6i). Altogether, iMG conditioned media was unable to recapitulate the significant neurodegeneration observed with direct iMG culture indicating that this phenotype may be more heavily dependent on direct neuron-microglia interactions rather than inflammatory mediator release. Altogether, this data demonstrate that FRDA microglia induce iron accumulation, ROS-dependent DNA damage and caspase-mediated apoptosis in healthy neurons through cell–cell mediated interactions. Importantly, addition of gene-edited microglia to healthy neurons prevented healthy neuronal stress and Caspase-3 upregulation.

## Xenotransplanted human FRDA microglia are hyperinflammatory

Given recent evidence of cerebellar microglia alterations in FRDA mouse models[18–20], we xenotransplanted human iPSC-derived hematopoietic progenitors (iHPCs) from a healthy donor, FRDA patient and an isogenic gene-corrected line into the Rag2$^{-/-}$Il2rg$^{-/-}$ CSF1$^{h/h}$ Csf1r$^{ΔFIRE/ΔFIRE}$ murine model. Of note, iHPCs did not exhibit cell line specific differences in expression of the yolk sac hematopoietic marker CD43, with equivalent differentiation potential (Supplementary Fig. 7a). This unique model is humanized for the essential microglial growth factor CSF1, amenable to cross species transplantation and is depleted of murine microglia via an enhancer deletion in the CSF1 receptor (fms intronic regulatory element)[56] to facilitate the complete engraftment of iHPCs as we have previously described[21] (Fig. 6a). Eight weeks post-transplantation, when mature microglia transcriptionally and epigenetically closely recapitulate primary human microglia[21], we assessed microglial density and distribution throughout the cerebellar cortex and found a robust enrichment of human microglia across the cerebellar lobules, as evidenced by the presence of the human nuclei

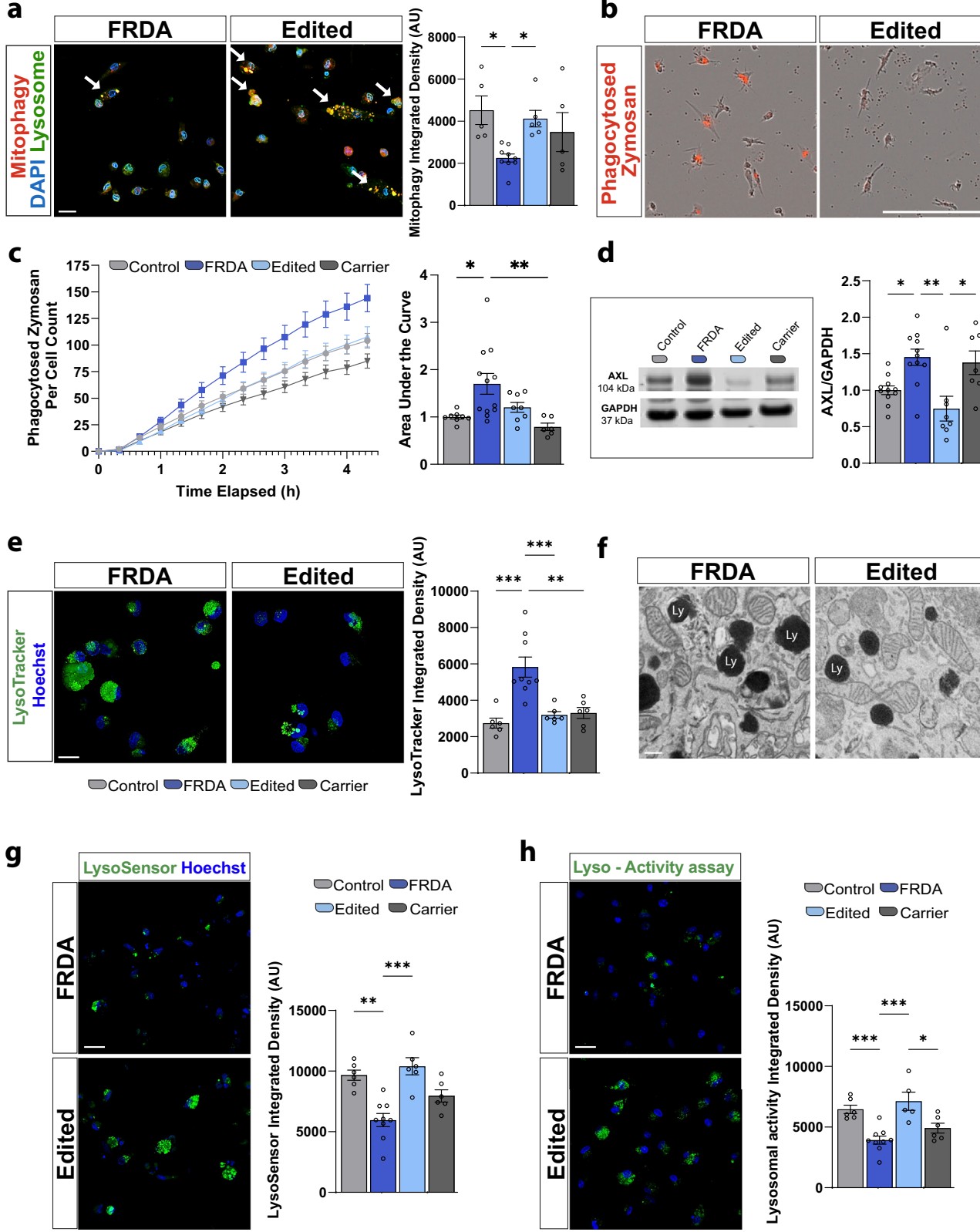

marker, Ku80+ and Iba1+ staining (Fig. 6b). Quantification of the microglia population across cerebellar layers revealed pronounced FRDA microglia accumulation in the Purkinje cell layer (PCL) and the white matter (WM), where Purkinje cell collaterals reside (Fig. 6b). To investigate the activation state of xMGs and correlate this with our in vitro findings, we performed a Sholl analysis of microglia morphology across cerebellar cortical layers from healthy, FRDA and gene-edited

xMGs. We identified an overall ameboid FRDA microglial morphology in the granule, Purkinje and molecular cell layer while the healthy and gene edited xMGs exhibited a homeostatic and ramified profile (Fig. 6c). We then quantitated the phagocytic microglial marker CD68, finding a marked accumulation of CD68+ puncta and increased CD68 puncta volume in FRDA xMGs compared to healthy and gene edited xMGs (Fig. 6d). Moreover, cerebellar FRDA xMGs exhibited increased

**Fig. 3 | FRDA microglia are hyperphagocytic with impaired lysosomal function.**
**a** Representative images (left) and quantitation (right) of mitophagy induced by three hours of 50 mM CCCP treatment. Note co-localization of the mitophagy dye (red) with the lysosomal dye (green) to confirm signal specificity. **b** Representative images of iMG phagocytosis of zymosan particles. **c** Time course of iMG phagocytosis of zymosan particles over 4.5 h (left) and quantitation of overall area under the curve (right). **d** Western blot (left) and quantitation (right) depicting increased expression of AXL protein in FRDA. **e** Confocal images (left) and quantitation (right) of the lysosomal marker, LysoTracker. **f** EM representative images (left) and quantitation (right) of lysosome numbers in control, FRDA and gene-edited lines. **g** Lysosomal pH dye (Lysosensor) representative (left) and quantitation (right).

**h** Representative images (left) and quantitation (right) of lysosomal activity assay. Each dot in the plot represents a biological replicate with $N = 3$ biologically independent replicates/differentiations for each iMG cell line. Two controls, three FRDA, two gene-edited and two carriers familial control lines for (**a**–**e**, **g**, **h**) and two controls, two FRDA and two gene-edited lines for (**f**). Integrated Density (AU: arbitrary units) has been quantified per each cell, with at least 80 iMG cells/line/experiment analyzed. For (**g**) two controls, two FRDA and two gene-edited lines are included and plotted in the analysis. Data are presented as Mean ± SEM, and groups analyzed by one-way ANOVA with Tukey's post-hoc correction. *$p < 0.05$, **$p < 0.01$, ***$p < 0.001$, ****$p < 0.0001$. Scale bars **a** 20 µm, **b** 200 µm, **e** 15 µm, **f** 40 pixels **g** 30 µm, **h** 30 µm.

expression of the MHC class II histocompatibility antigen human leukocyte antigen-DR (HLA-DR) and concurrent loss of the homeostatic marker P2YR12, in the cerebellum and in the brain stem of adult mice, further reinforcing the pathological characteristics of these diseased microglia (Fig. 6e and Supplementary Fig. 7b). Furthermore, we then used 8-Hydroxydeoxyguanosine (8-OHdG) as a marker for oxidative stress to detect oxidative DNA damage as a consequence of ROS accumulation in xMGs. Importantly, we demonstrated that xenotransplanted FRDA xMGs exhibit elevated oxidative stress, as evidenced by a higher proportion of 8-OHdG–positive FRDA xMGs (Fig. 6f). Of high relevance, xenotransplantation of gene-corrected iHPCs prevented the inflammatory phenotype of FRDA xMGs, closely mimicking our in vitro findings.

### Human FRDA microglia disrupt the survival of healthy Purkinje cells in vivo

Given that marked cerebellar degeneration is a hallmark of FRDA pathology, we next investigated whether xenotransplant of FRDA microglia progenitor cells could recapitulate the cerebellar neurodegenerative phenotype observed in primary human FRDA tissue and in FRDA mouse models. As Purkinje cell degeneration has been recently reported in an FXN humanized FRDA mouse model[20], we queried whether FRDA xMGs perturb Purkinje cell survival. We used the calcium-binding protein Calbindin-D28K and Inositol-1,4,5 triphosphate receptors (IP3Rs), as Purkinje cell markers and found a significant reduction in Purkinje cell neurons in adult mice xenotransplanted with FRDA xMGs when compared to untransplanted mice and mice xenotransplanted with healthy and gene-edited xMGs (Fig. 7a and Supplementary Fig 7c). Due to the elevated accumulation of ROS-induced DNA damage in healthy neurons co-cultured with FRDA iMGs (Fig. 5f, g), we questioned whether a similar mechanism might occur in vivo and found that Purkinje neurons in mice xenotransplanted with FRDA xMGs were strongly enriched in 8-OHdG (Fig. 7b), suggesting that oxidative stress could play a significant role in the pathophysiology of Purkinje cell injury. Microglia are believed to contribute to the elimination of immature Purkinje cells during cerebellar development[57]. We therefore investigated whether abnormal microglia activity during cerebellar development could account for the observed Purkinje cell loss. At postnatal day 21 (P21), immediately after the closure of the major phase of cerebellar development, we found comparable numbers of Purkinje cells between xenotransplanted groups (Supplementary Fig 7d), indicating that Purkinje cell loss occurred in adulthood rather than during development. Finally, we investigated whether the neurodegenerative phenotype observed in FRDA xMGs mice was restricted to Purkinje cells. By using NeuN as a neuronal marker, we assessed neuronal density in the granule cell layer of adult mice and showed a similar impact of FRDA xMGs on the density of granule cell neurons (Fig. 7c). Altogether, these findings support the pathological role of FRDA microglia in shaping the cerebellar cortex of adult mice, leading to cerebellar neuronal degeneration. Importantly, gene editing effectively mitigated this neurodegenerative phenotype, achieving neuronal numbers comparable to those observed in healthy control lines.

## Discussion

FRDA is a multisystemic, devastating, progressive neurodegenerative disease with no cure currently available. The only current FDA-approved therapy is omaveloxolone, which reduces inflammation and oxidative stress via induction of the transcription factor NRF2[58]. FRDA patients have a broad clinical phenotype with marked neuronal degeneration in the spinal cord, dorsal root ganglia and dentate nucleus neurons, together with marked alterations in cerebellar Purkinje cells[11,12,59]. More recently, microglia dysfunction has been reported in post-mortem FRDA human samples and in different models mimicking the pathology. Microglia alterations in postmortem human samples include marked microglia activation, elevation of ferritin⁺ ameboid microglia cells and elevation in inflammatory cytokines[17]. In multiple murine models of FRDA, microglial dysfunction has also been noted. This includes early cerebellar microglial activation with a less ramified microglia morphology, changes in microglial gene expression in pathways that regulate inflammation and oxidative stress, a metabolic shift from OXPHOS to elevated glycolysis, increased phagocytosis, impaired migratory capacity, and overall disrupted microglia homeostasis[9,10,18,19,60]. Additionally, an increase in mRNA of key proinflammatory cytokines has been linked to cerebellar atrophy with loss of Purkinje cells[20]. However, whether microglial dysfunction is a primary pathogenic driver of FRDA pathology or a secondary reaction to neuronal dysfunction, apoptosis, and death cannot be discerned from murine models where the GAA repeats are ubiquitously present in all cell types.

Here, we provide evidence that microglial dysfunction is a significant contributor and could be an independent mediator of FRDA pathology. In postmortem human cerebellar sections, we find that cerebellar microglia from FRDA patients display reduced ramification complexity, reflective of a hyperactivated phenotype. Next, using a robust cohort of human iPSC-derived microglia, we were able to dissect the disease-specific phenotype of FRDA microglia in vitro. We showed that FXN loss in microglia leads to a cell-autonomous ameboid morphology and increased expression of proinflammatory markers, such as CD68 and Iba1. These findings align with previous observations of microglia hyperactivation in FRDA pathology and suggest that microglia may actively contribute to the neurodegenerative FRDA phenotype[19,61]. Of note, the restoration of FXN protein expression in gene-edited microglia, achieved by excision of GAA repeats in the FXN gene, attenuated this hyperinflammatory microglia phenotype, highlighting the potential of CRISPR/Cas9-based therapies to target microglia activation in FRDA pathology.

The pathological impact of FXN loss in microglia extends beyond inflammation, affecting key cellular processes that alter cellular homeostasis. Building on prior evidence of mitochondrial dysfunction in other FRDA cell types[3,7,28] these findings demonstrate similar effects in human microglia cells. We showed that FRDA microglia exhibit pronounced mitochondrial dysfunction, including increased mitochondrial oxidative stress, compromised mitochondrial membrane potential, and disrupted mitochondrial morphology, with a higher incidence of spherical and possibly altered mitochondrial cristae structure. Deficits in mitochondrial respiration were associated with

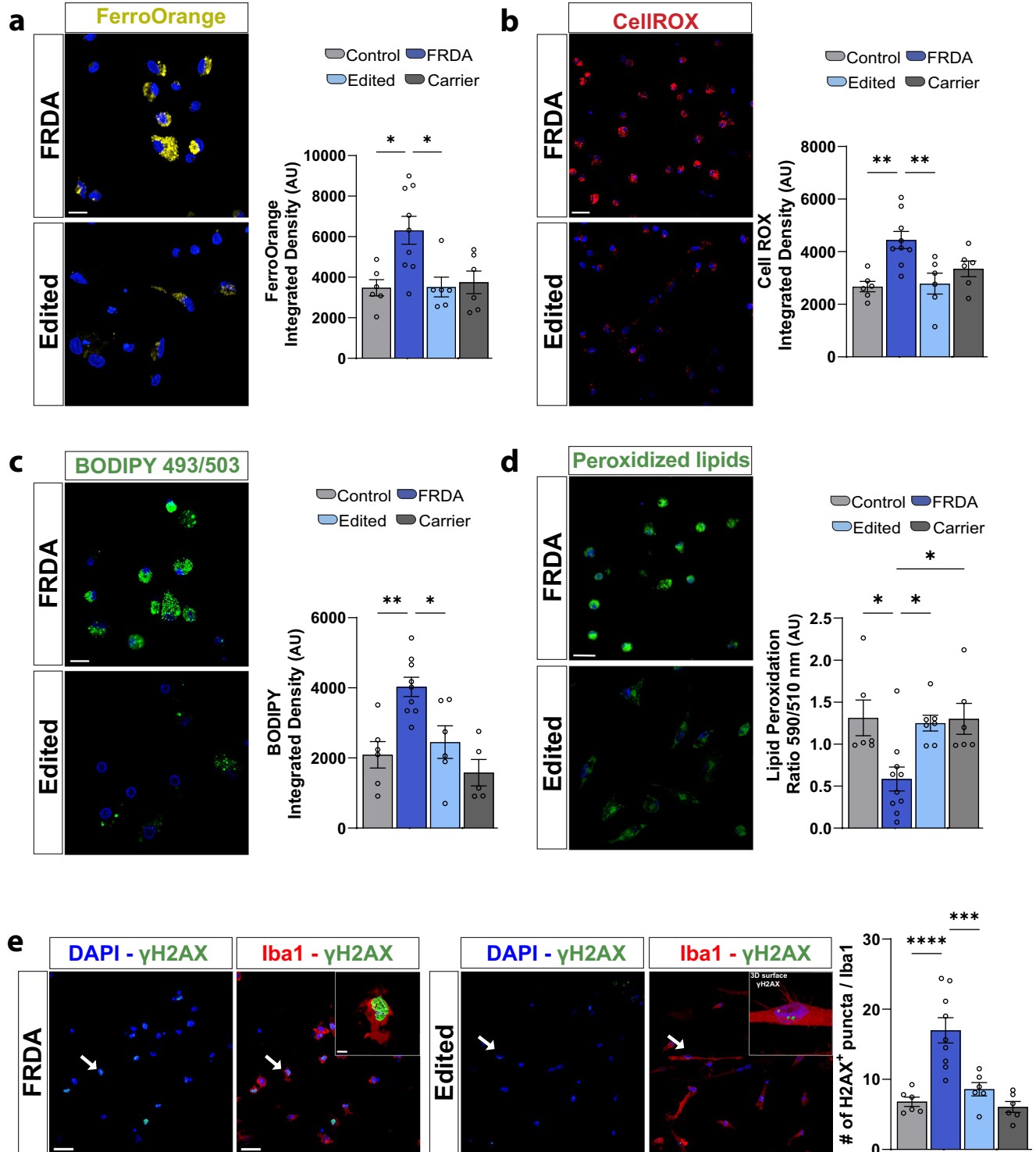

**Fig. 4 | FRDA microglia have disturbed cellular homeostasis corrected by gene editing. a** Live cell imaging of FerroOrange dye used to assess iron levels within iMG (left) with quantitation (right). **b** CellRox levels in iMG (left) and quantitation (right) across genotypes. **c** Live cell imaging of neutral lipids (BODIPY493/503) (left) and quantification (right). **d** Live cell imaging of lipid peroxidation (left) and quantification (right). Ratio is total lipids (590 nm)/peroxidized lipids (510 nm). **e** Confocal images of γH2AX (left) with quantitation of γH2AX+ puncta/cell using IMARIS 3D surface reconstruction. Each dot in the plot represents a biological replicate with $N = 3$ biologically independent replicates/differentiations for each iMG cell line. Two controls, three FRDA, two gene-edited and two familial carrier lines for (**a**–**c** and **e**); Two controls, four FRDA, three gene-edited and two familial carrier lines for (**d**). Integrated Density (AU: arbitrary units) has been quantified per each cell, with at least 80 iMG cells/line/experiment analyzed. Data are presented as Mean ± SEM and analyzed by One-way ANOVA with Tukey's post-hoc testing. *$p < 0.05$, **$p < 0.01$, ***$p < 0.001$, ****$p < 0.0001$. Scale bars **a** 10 µm, **b** 30 µm, **c** 20 µm, **d** 20 µm, **e** 30 µm and 5 µm (3D puncta reconstruction).

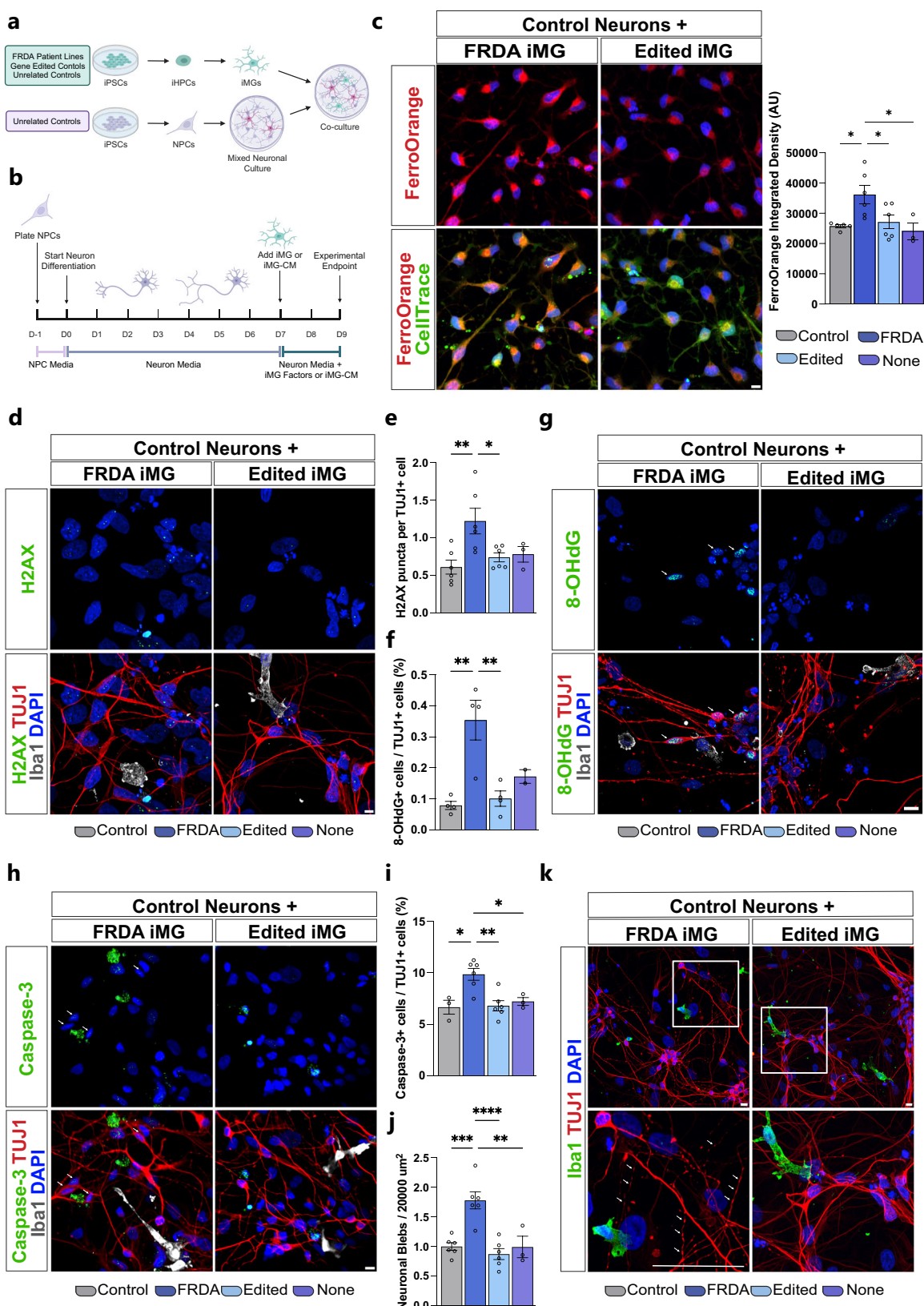

reduced ATP production and increased ROS accumulation, findings in agreement with the well-documented role of frataxin in mitochondrial iron-sulfur cluster biosynthesis and OXPHOS[2,4]. Surprisingly, familial carriers who are heterozygous for the GAA repeat in a single allele and show no clinical phenotype exhibited some mild microglia impairments. Whether these subtle microglial deficits are sufficient to cause

premature microglial aging or contribute to late-onset cognitive or other subtle deficits remains to be explored.

Alongside mitochondrial defects, we also reported marked lysosomal dysfunction in FRDA. These abnormalities include higher lysosomal density, altered lysosomal pH, and attenuated lysosomal activity. These defects might explain the reduced mitophagy efficacy, a

**Fig. 5 | FRDA microglia induce neuronal degeneration via caspase 3-mediated apoptosis. a** Schematic of microglia, neuron and astrocyte co-culture created *in BioRender. Coufal, N. (2025)* https://BioRender.com/84udn2k. **b** Timeline of co-culture and conditioned media experiments created *in BioRender. Coufal, N. (2025)* https://BioRender.com/ii1sd2x **c** FerroOrange staining for intracellular iron (left) and quantitation (right) (AU: arbitrary units) in healthy neurons co-cultured with control, FRDA or gene-edited iMG. Neurons labeled with CellTrace CFSE dye prior to co-culture to differentiate from iMG. **d**, **e** Immunostaining (left) and quantification (right) of γH2AX for foci of DNA damage in healthy TUJ1+ neurons co-cultured with iMGs. **f**, **g** Immunostaining (right) and quantification (left) of healthy TUJ1+ neurons positive for 8-OHdG after co-culture with iMGs. Arrows indicate

8-OHdG+ TUJ1+ cells. **h**, **i** Immunostaining (left) and quantification (right) of cleaved caspase-3 in healthy TUJ1+ neurons co-cultured with iMGs. Arrows indicate caspase-3+ TUJ1+ neurons. **j**, **k** Immunostaining for TUJ1 (right) and quantification (left) of neuronal blebbing in healthy neurons co-cultured with iMGs, arrows indicate exemplar blebs. Data normalized to control. *n* = 2 FRDA, 2 edited, and 2 non-related iMG controls. Neurons from 1 healthy donor line with 3 biologically independent replicates/differentiations for each iMG cell line. Data are represented as mean ± SEM and analyzed by one-way ANOVA with Tukey's posthoc test where *\*p* < 0.05, \*\**p* < 0.01, \*\*\**p* < 0.001, \*\*\*\**p* < 0.0001. Scale bars **c** 20 μm, **d** 15 μm, **g** 5 μm, **h** 10 μm, **k** 15 μm (magnification 0.05 in.).

lysosome-dependent type of autophagy, and subsequent accumulation of dysfunctional organelles within FRDA iMGs. Attenuation of mitophagy was also accompanied by a hyper-phagocytic profile of FRDA microglia in vitro, a finding that was also supported by the increased expression of AXL, a phagocytic microglial marker. These data suggest that the accumulation of dysfunctional organelles within cells might contribute to the hyperinflammatory profile of FRDA iMGs in vitro. All of the above-mentioned organelle dysfunctions can lead to a vicious cycle of oxidative stress and cellular damage within cells[41–48]. Here, we reported that the lack of FXN protein in iPSC-derived FRDA microglia leads to excessive iron overload, perturbed lipid peroxidation and marked DNA damage. Activation of this cascade strongly regulates neuroglial interactions. To explore how this pathological profile of FRDA microglia affects neuronal survival, we assessed FRDA microglial interaction with healthy neurons in a 2D in vitro co-culture model, and we showed that FRDA microglia promote neuronal iron accumulation, lead to ROS-dependent DNA damage, induce membrane blebbing and upregulate the expression of the cysteine-proteases Caspase-3 in healthy neurons. This result suggests that microglial dysfunction in FRDA may not only contribute to local inflammation but could also lead to healthy neuronal degeneration, a phenotype that is successfully prevented by GAA correction in gene-edited iMGs. Xenotransplantation of FRDA microglia progenitor cells (xMGs) into a humanized mouse model recapitulated the hyperinflammatory microglia profile we described in post-mortem human FRDA patients and in iPSCs-derived microglia. In contrast to healthy controls and gene-edited lines.

xMGs, FRDA microglia appear ameboid in their morphology, upregulate the phagocytic marker CD68, lose the expression of homeostatic marker P2YR12 with subsequent upregulation of the inflammatory antigen presentation marker HLA-DR and are enriched with 8OHdG expression, a well-known marker for ROS-induced DNA damage. FRDA xMGs were found to accumulate in the Purkinje cell layer of the cerebellum. Purkinje cells exhibited abundant 8-OHdG accumulation within their soma with an overall marked reduction in Purkinje cell number only in adulthood and not in juvenile xenotransplanted mice, suggestive of substantial Purkinje neuron degeneration in otherwise control mice, possibly mediated by ROS-dependent DNA damage.

Neurodegeneration was not limited to Purkinje cells but also affected the survival of cerebellar granule neurons, indicating a broader cerebellar neurodegenerative phenotype in FRDA. This observation is consistent with recent findings in the FRDA mouse models[18–20] and with human magnetic resonance imaging studies showing gray matter loss in the cerebellum of FRDA patients[62,63]. Importantly, our gene editing approach prevented the hyperinflammatory phenotype of xMGs in adult mice and preserved neuronal density to a comparable level as mice transplanted with healthy donor controls. This finding underscores the neurotoxic potential of FRDA microglia in the cerebellum, a region particularly vulnerable in FRDA patients, and highlights the ability of FRDA microglia to induce disease-specific neuropathology in vivo. Even though neurons are particularly susceptible to the accumulation of 8-OHdG[64], which could

potentially drive degeneration, a limitation of the current study includes the lack of a clear understanding of the mechanism driving microglial-induced healthy neuronal stress and degeneration. In our experimental approach, we found that healthy neuronal stress and degeneration appear to be cell-interaction dependent and not only through diffusible cytokine or other inflammatory mediators. Additionally, future investigation of behavioral phenotypes, which arise late in FRDA murine models (>8 months)[65] and therefore will be difficult to assess in immunocompromised mice with a shorter lifespan, is needed. Lastly, the interaction between neuroinflammation, neurodegeneration, and inflammation or immune activation is a key area not addressed within these studies.

Furthermore, these studies support the therapeutic approach under development, which involves autologous transplantation of gene-edited hematopoietic stem and progenitor cells (HSPCs). We previously demonstrated that a bone marrow transplant of wild-type murine HSPCs into young FRDA model mice prevented development of FRDA pathology, including neurodegeneration, and engrafted to some extent in the CNS differentiating into brain myeloid cells[66]. Given that FXN overexpression is toxic, we developed a strategy to remove the GAA expansion mutation using CRISPR/Cas9 technology and dual guide RNA, resulting in increased FXN expression in FRDA patients' CD34+ HSPCs[12]. In this present study, we show that FXN restoration via the same CRISPR/Cas9 gene editing approach was able to reverse many of the pathological features associated with FXN deficiency in FRDA patient-derived microglia, including mitochondrial and lysosomal abnormalities, possibly leading to attenuation of their inflammatory profile, which prevented neuronal death in vitro. Similarly, gene-edited FRDA microglia showed a reduced inflammatory profile and prevented Purkinje cell loss in vivo, suggesting that targeted gene editing may offer a therapeutic strategy to mitigate microglial-driven neurodegeneration in FRDA pathology. While the primary focus in FRDA has traditionally been on neuronal dysfunction due to FXN deficiency, our data suggest that microglia may play a crucial role and could be an important independent mediator in disease progression. Together, these findings support that microglial replacement in the CNS should have therapeutic efficacy in FRDA. These findings not only enhance our understanding of the cellular mechanisms underlying FRDA pathology but also support the clinical translation of the gene-edited HSPC therapy for FRDA.

## Methods
### FRDA patient cerebellum
Formalin-fixed post-mortem human cerebellar tissue was obtained from the University of Maryland Brain Bank, a member of the NIH NeuroBiobank project. Patient or parents consented within the NIH NeuroBioBank program, and deidentified samples, exempt from local IRB review, were provided for the study. Tissue samples were obtained from individuals with Friedreich's ataxia (FRDA) and from healthy donors. The cohort included three male and four female participants, spanning a wide age range from 3 to 64 years old. Six of White ethnicity, and one of Black African ancestry. No clinical diagnosis was reported for the healthy donors.

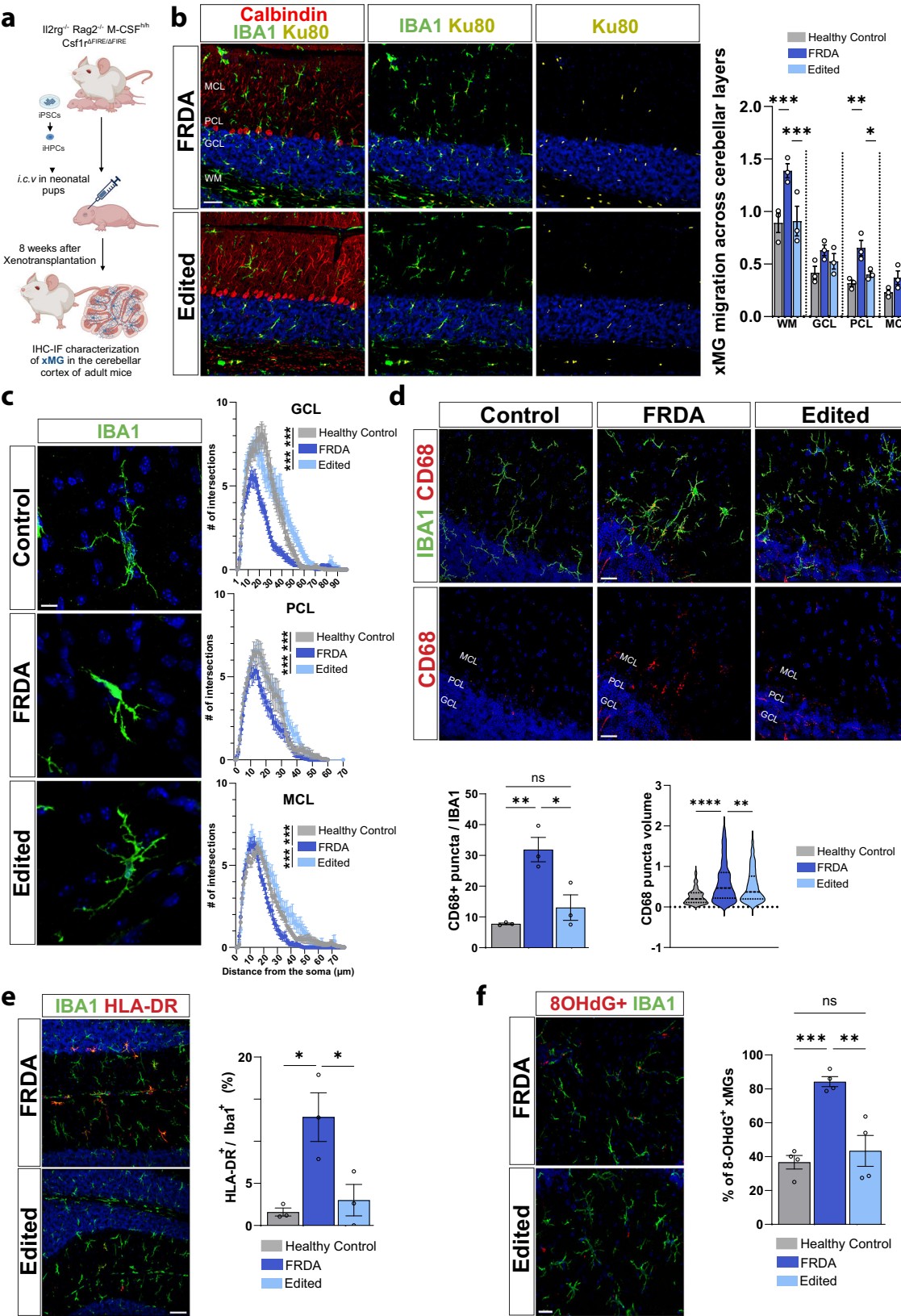

## Cell lines

In this study, we utilized iPSCs derived from three FRDA patients, two of which have been CRISPR/Cas9 GAA repeat depleted to generate respective isogenic controls, two familial carriers and two unrelated healthy donors as previously described[8]. FRDA iPSCs lines were generated from fibroblasts derived from the FF1 (GM03816), FF2

(GM23913) and the 850 (GM15850) patient lines purchased from Coriell Institute. 223 FRDA (GM16223) and 264 (GM22264) unrelated healthy control lines were generated from patients' lymphoblast cell lines purchased from Coriell Institute. A previously established healthy control iPSC line was also utilized. This cohort has been previously described[8], in addition two new familial carriers include 236

**Fig. 6 | Human FRDA microglia xenotransplanted in a non-FRDA murine are hyperinflammatory. a** Schematic overview of xenotransplants created *in BioRender. Coufal, N. (2025)* https://BioRender.com/c83g767. **b** Cerebellum of adult FIRE mice xenotransplanted with hematopoietic progenitors from one healthy, one FRDA and one gene-edited patient lines labeled with Iba1 (microglia marker), Ku80 (human nuclei marker), Calbindin (PC marker) and DAPI (nuclei) (left) with quantitation by cerebellar layer (right). *N* = 3 mice per xenotransplanted genotype. **c** Confocal images of cerebellar xMGs stained with IBA1. Sholl analysis of xMGs for microglia morphology in the granule cell layer (GCL), molecular cell layer (MCL) and Purkinje cell layer (PCL). *N* = 3 mice per xenotransplanted genotype. **d** Representative pictures of CD68 in cerebellar microglia of adult FIRE mice with

quantitation for CD68+ puncta number/IBA1 and puncta volume in healthy, FRDA and gene edited transplanted mice *N* = 3 mice per xenotransplanted genotype. **e** Representative confocal images for Iba1 and human leukocyte antigen-DR (HLA-DR) expression in the cerebellar layers of adult mice. *N* = 3 mice per xeno-transplanted genotype. **f** Representative confocal images and quantification for Iba1 and 8-Hydroxydeoxyguanosine (8-OHdG) in xMGs. *N* = 4 mice per xeno-transplanted genotype. One cell line/genotype. Data are presented as mean ± SEM and analyzed by one-way ANOVA, with Tukey's post-hoc testing. *$p < 0.05$, **$p < 0.01$, ***$p < 0.001$, ****$p < 0.0001$. Scale bars **b** 100 μm, **c** 10 μm, **d** 20 μm, **e** 100 μm, **f** 30 μm.

(GM16236) and 849 (GM15849) cell lines were generated. 223 FRDA, 850 FRDA patients LCLs and FF2 iPSCs lines underwent gene editing for *FXN* pathological expansion mutation utilizing our established CRISPR/Cas9 protocol[8]. All the overmentioned lines will be referred to as FF1, FF2 and 223 FRDA; FF2 and 223 Edited; 236 and 849 Carrier; and 264 and EC11 healthy controls. Studies were conducted under a protocol approved by the Institutional Review Board of the University of California, San Diego (UCSD) under IRB 171379.

## Reprogramming of patient lymphoblast cell lines and iPSCs generation

Deidentified patient lymphoblast cell lines (LCLs) were obtained from Coriell Cell Repository, a cell resource and biorepository that includes consent for reprogramming. LCLs were reprogrammed using the Epi5 Episomal iPSC Reprogramming Kit (Life Tech, #A15960) under IRB 17379. Cells were counted and resuspended in Nucleofector Solution to prime cells for nucleofection. The cell suspension was combined with either pmax GFP Vector to evaluate nucleofection efficiency or Epi5 reprogramming vectors and EBNA to reprogram the cells to iPSCs. After completing the appropriate program on the Nucleofector, cell solution was added to pre-equilibrated N2 and B27 (Thermo-Fisher) media and seeded on Cultrex-coated six well plates. iPSCs were then cultured in mTESR™ Plus (StemCell Technologies) coated with Cultrex Basement Membrane Matrix (Trevigen) at a concentration of 1 mg/24 mL. A CRISPR/Cas9 approach was applied to gene-correct iPSCs, which give rise to a mixed population of cells. Using TrypLE media, cells were dissociated into single cells and plated on a 96-well plate previously coated with Cultrex. Colonies were grown at a desired size and isolated using Collagenase IV and subsequently screened to evaluate the success of the CRISPR/Cas9 approach using PCR. Selected clones were validated with a DNA fingerprinting test for normal karyotyping.

## PCR for GAA repeat excision

PCR for GAA repeats was performed as previously described[8]. Briefly, QuickExtract kit (Lucigen) was used to extract genomic DNA from FRDA gene-edited, carrier and control iPSCs. Gene Editing (GE) fwd: 5′-GGT GTA GGA TTA AAT GGG AAT AA-3′; Rev: 5′ GGA TGC ACA GGA GCT TATT-3′, and Non-Gene Editing (NGE) fwd: 5′-GGA CCT GGT GTG AGG ATT AAA-3′; rev: 5′-CTA ATA CAT GCG GCG TAC CA-3′ primers were used to test the efficacy of our CRISP/Cas9 approach and to select the clones that have been used in this study. PCR reaction was carried out with GoTaq Green Master Mix 2X (Promega) and the following settings were used: 95 °C for 3 min (95 °C for 30 s, 52 °C (GE) and 56 °C (NGE) for 30 s, 72 °C for 1 min) × 35, 72 °C for 10 min. PCR products were loaded and then run on a 1% agarose gel at 120 V for 30 min. GE and NGE bands were detected at 401 bp and 313 bp, respectively, using a 1 Kb DNA ladder.

## Generation of iHPCs and iMG differentiation

Induced hematopoietic progenitor cells (iHPCs) were generated using the StemDiff HPC differentiation kit (StemCell Technologies) following manufacturer instructions. Briefly, iPSCs were enzymatically treated

with ReLeSR (StemCell Technologies) and plated at a low density in mTESR™ Plus media containing 1 μM Rock inhibitor (Fisher Scientific, #HY-10583-10MG). The following day, the media was changed to STEMdiff Media A (STEMdiff Hematopoietic Basal Medium with STEMdiff Hematopoietic Supplement A) at a 1:200 dilution, and on Day 4, the media was changed to STEMdiff Media B (STEMdiff Hemato-poietic Basal Medium with Hematopoietic Supplement B) at a 1:200 dilution. On days 11–13, non-adherent cells were collected, resuspended in DPBS (Gibco), and seeded at approximately $2 \times 10^5$ on a 6-well plate for differentiation in microglia complete media which consists of DMEM F12 (Gibco) with 1X Insulin-Transferrin-Selenium (Gibco), 2% B27 Supplement (Gibco), 0.5% N2 Supplement (Gibco), 400uM monothioglycerol, 1X Glutamax (Gibco), 1X MEM NEAA (Gibco), 2.5 μg/mL Insulin (Sigma), 25 ng/ml M-CSF (Humanzyme, #HZ-1039), 100 ng/ml IL-34 (Humanzyme, #HZ7064), and 50 ng/ml TGFb-1 (Humanzyme, #HZ-1087). For the next 28 days, 1 mL of microglia complete media was added per well every other day. iMGs were mature after 28 days in complete media.

## Microglia Immunofluorescence

Immunofluorescence was performed as previously described. Briefly, iMGs derived from FRDA, edited, carriers and controls line were seeded at a density of $6 \times 10^4$ cells per slide in 8-well bottom glass slides (Millipore) for fixed stains, and/or on Ibidi glass bottom 8-well slides (Ibidi) for live cell imaging, previously coated with CellTak at a concentration of 1:200 in ultra-pure $H_2O$. The following day, coverslips were washed in PBS and fixed using 4% paraformaldehyde (EMS, #19210) for 20 min. After fixation, cells were washed in PBS and then blocked for 60 min in 10% normal donkey serum (NDS) + 0.5% Triton-X100 at room temperature (RT). Cells were then incubated in primary antibodies diluted in 3% NDS + 0.5% Triton-X100 overnight at 4 °C. The next day, cells were rinsed in PBS and incubated with the appropriate secondary antibody 1 at RT with gentle agitation. Cells were incubated with 1 μl/mL of DAPI (ThermoFisher, #D1306) for 10 min and mounted for imaging with Immu-Mount (ThermoFisher, #9990402).

## Western blot

1 million iMGs were lysed in 1X RIPA buffer (Millipore, #20-188) supplemented with 1X protease/phosphatase inhibitors (ThermoFisher Scientific, #78443). Lysate was run on 4–12% NuPAGE acrylamide gels (Invitrogen, #NP0321BOX) in 1X MOPS buffer (Invitrogen, #NP0001) for 20 min at 150 V followed by 45 min at 180 V and transferred to 0.22 μm nitrocellulose blotting membrane (ThermoFisher Scientific, #IB23002). Blocking was performed with 4% milk (Cell Signaling Technologies, #9999S) for 1 h. Membrane was incubated with primary antibodies overnight at 4 °C. Primary antibodies included Frataxin, TOM20, OXPHOS, AXL, GAPDH, LAMP1. The next day, IgG HRP-linked secondary antibody was added to visualize FXN. Pierce Super Sensitivity ECL Western Blotting Substrate (ThermoFisher Scientific, A38554) was used to visualize protein bands in a ChemiDoc MP Imaging System 1341 (BioRad). Alexa Fluor Plus 680 and 800 were used to visualize OXPHOS, AXL and LAMP1 using a Licor detection approach.

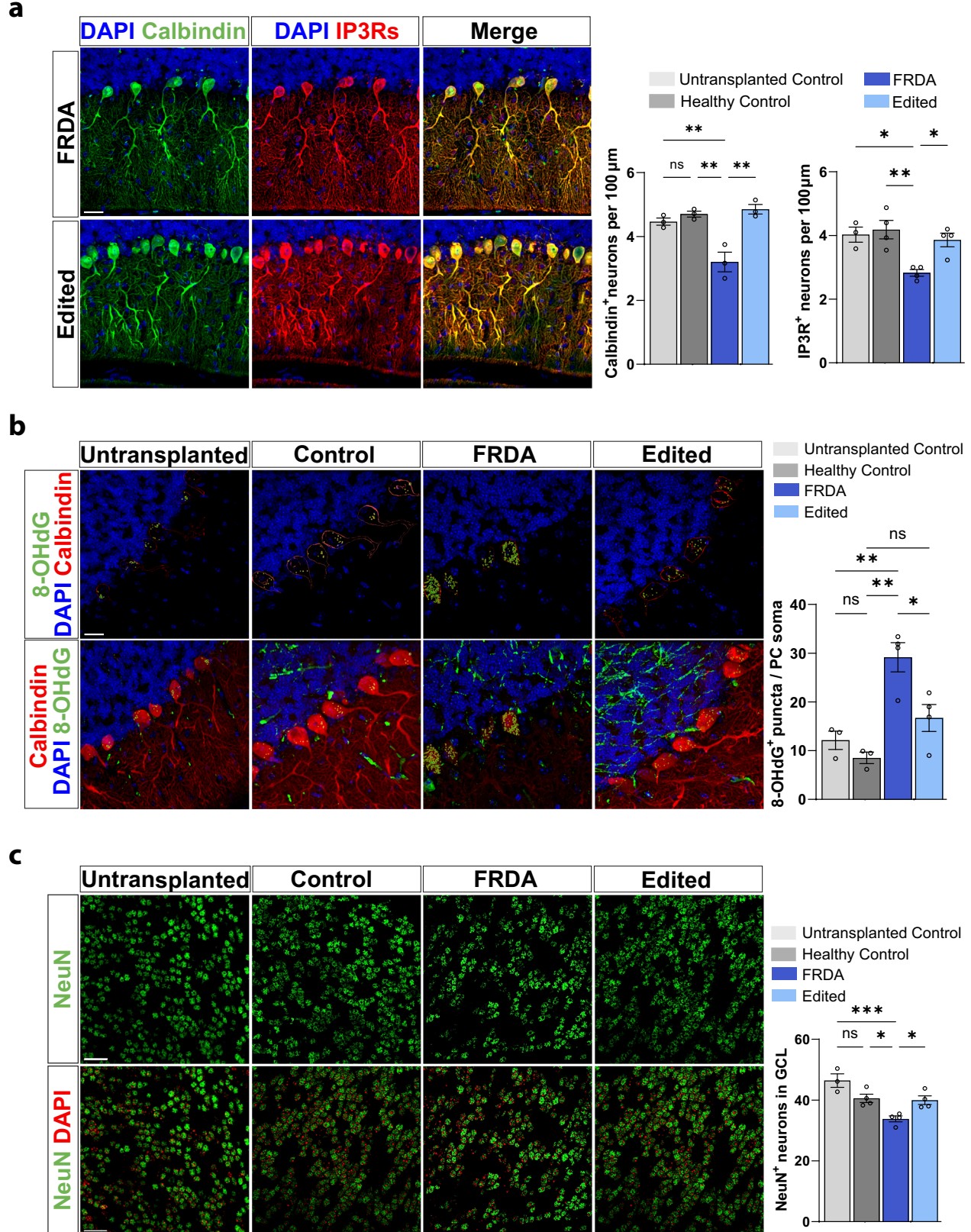

## Flow cytometry

iMGs were washed once in PBS (Gibco, #14190-144), spun down, and then resuspended in FACS sorting buffer: HBSS (Gibco, #14175-095), 1% BSA (Sigma-Aldrich, #A9647-10G), 1 mM EDTA (Sigma-Aldrich, #E-7889). The cell suspension was blocked in 1:20 human TruStain FCX (BioLegend, #422302) for five minutes at room temperature. Cell suspension was incubated for 20 min on ice using 1:200 fluorophore-conjugated antibodies: CD43, CX3CR1, CD11b, CD68 and CD45. From each sample, 10 µL of the blocked cell suspension was collected prior to antibody incubation and pooled for an isotype control. Isotype control antibodies were APC, PerCP-Cy5.5, 488 FITC, PE, PE-Cy7 and APC Cy7 were added to the isotype control sample. Zombie violet was

**Fig. 7 | Human FRDA microglia xenotransplanted in healthy mice induce cerebellar neuronal death. a** Calbindin (PC marker) and IP3Rs (Inositol-triphosphate receptor 3) were used to quantify Purkinje cell density in the cerebellar cortex of adult mice. $N = 3$ mice per xenotransplanted genotype for Calbindin quantification and $N = 4$ mice per xenotransplanted genotype for IP3R quantification. **b** Representative confocal images of 8-OHdG as a ROS-induced DNA damage marker in Purkinje cells stained with Calbindin. 3D puncta reconstruction highlighted in yellow (top) and merged staining (bottom). $N = 4$ mice per xenotransplanted genotype. **c** NeuN staining in the cerebellar cortex of adult FIRE mice used to assess neuronal loss in the cerebellar granule cell layer from 10 random ROIs of 2500 $\mu m^2$ for each mouse is included in the analysis. $N = 4$ mice per transplanted genotype. $N = 3$ for untransplanted control in (**a–c**). One cell line/genotype. Data are presented as mean ± SEM and analyzed by one-way ANOVA, with Tukey's post-hoc testing. *$p < 0.05$, **$p < 0.01$, ***$p < 0.001$, ****$p < 0.0001$. Scale bars **a** 30 $\mu m$, **b** 20 $\mu m$, **c** 50 $\mu m$.

added at a concentration of 1 µg/mL to determine cell viability. The sample was then washed and resuspended in FACS buffer.

### Ramification quantification in vitro

Characterization of iMG ramifications was carried out using an Incu-CyteS3 Live-Cell Analysis System (Sartorius), a high-throughput image analysis software. Microglia were plated at a density of 10k cells per well in a 96-well Primaria-treated plate (Corning, #353872). Phase images of iMGs were obtained at 10×, and the IncuCyte NeuroTrack Analysis Software module was utilized for automated quantification of ramification morphology. First, cell bodies were segmented from the background based on texture and brightness and were masked as cell body clusters. Second, linear features were detected based on brightness and width and were masked as ramifications. Ramification length (mm) per cell body cluster and ramification branch points per cell body cluster were quantified and plotted using GraphPad Prism 10.

### Zymosan phagocytosis

Microglia were plated on 96-well Primaria plates (Corning, #353872) 1 day before the assay and were labeled with lectin (Invitrogen, #L32470) at a concentration of 1:1000. Zymosan beads conjugated to pHrodo (ThermoFisher, # P35364) were spiked in at a concentration of 1:100. Plate was imaged on IncucyteS3 Live-Cell Analysis System every 20 min for 4 h.

### Live cell imaging

To detect mitochondrial superoxides, cells were incubated in 1 uM of MitoSOX Red and 1 nM MitoTracker Green FM for 15 min at 37 °C. To detect mitochondrial membrane potential, iMGs were incubated with MitoTracker Deep Red for 30 min at 37 °C. Lysosomal characterization has been carried out with LysoTracker, LysoSensor and the lysosomal activity assay following manufacturer's instructions. To measure mitophagy, cells were stained with mitophagy dye for 30 min followed by treatment with CCCP at 50 mM for 3 h. Iron accumulation was measured using FerroOrange, and neutral lipids were quantified using BODIPY 493/503. Lipid peroxidation was detected using the Image-iT Lipid Peroxidation kit following the manufacturer's instructions. After incubation with the appropriate dyes, cells were washed twice with dPBS and incubated in PBS containing Hoechst at a concentration of 1:2000 for 10 min at room temperature, washed with PBS and imaged live using Andor Dragonfly spinning confocal microscope.

### Electron microscopy

Mature iMGs were cultured as described above and plated in a Cell-Tak (Corning, #354240) coated Ibidi 8-well #1.5 glass coverslip culture slides (Ibidi, #80826) with a density of 50k/well. All supplies for electron microscopy were sourced from Electron Microscopy Sciences (Hatfield, PA) unless noted otherwise. Cell media was carefully aspirated with a micropipette, leaving a minimal layer covering the cells. Fixative of 3% glutaraldehyde, 4% paraformaldehyde, 3 mM calcium chloride, and 0.1 M sodium cacodylate (pH 7.4) prewarmed to 37 °C was added to fill each well. Within 5 min of initial fixation, wells were carefully exchanged with ice-cold fixative and left at 4 °C in the dark for 90 min. Cells were rinsed 5 × 3 min with ice-cold 3 mM calcium chloride in 0.1 M sodium cacodylate (cacodylate buffer). The buffer was then exchanged with reduced osmium (1.5% osmium tetroxide, 1%

potassium ferrocyanide, 3 mM calcium chloride, 0.1 M sodium cacodylate) and was left to post-fix and stain in the dark at room temperature for 45 min. The reduced osmium was exchanged with ice-cold buffer rinses 5 × 3 min. Samples were stained further with filtered 1% aqueous uranyl acetate for an hour at room temperature before serial dehydration with ice-cold changes of ascending ethanol concentrations (30/50/70/80/90/100/100%) for 10 min each. After two rinses with anhydrous ethanol at room temperature, ethanol was exchanged for a 1:1 mixture of ethanol with resin (Eponate 12 Hard formulation, Pelco). Samples were left covered on a rocking device for 3 h. The slide was inverted over a waste container, and the ethanol-resin mixture was exchanged with about 0.5 mL of pure resin per well. This process was repeated three times with 30-min intervals on the rocking device. The sample wells were then filled completely with fresh resin and placed in a 70 °C oven for 20 h to polymerize. The chambered slide was then carefully plunged between a beaker of liquid nitrogen and a beaker of boiling water until the glass coverslip popped off, leaving the resin-embedded cells exposed and supported by the Ibidi well-chambers. The sample was placed cell-side up in a small miter box, and "wafers" of the resin-embedded wells (about 3–5 mm in width) were cut with a fine kerf saw. Wafers from the same well were collected into labeled bead bags until further processing. Wafers were then loaded into a small vice, and small blocks were cut to fit a vice sample holder on a Leica UC7 ultramicrotome for *en face* sectioning. Block faces of the exposed cells were trimmed with a razor blade to a dimension of about 2 × 3 mm. Ultrathin sections (70–80 nm) from the block face were cut using a Diatome Histo diamond knife and collected onto diced ultra-flat silicon wafer chips (University Wafers, Boston, MA). These were briefly dried on a hotplate set to 60 °C, labeled with a diamond scribe, and secured to aluminum stubs using sticky carbon tabs. The assembled samples were then loaded into the scanning electron microscope (SEM; Zeiss Sigma VP) for imaging. Images were collected using a Gatan 3VBSD2 backscattered electron detector at a working distance of approximately 6 mm, with accelerating voltage set to 3 kV in high current mode and using a 30 µm aperture. The array tomography module in Atlas5 control software (FIBICS) was used to generate a low-resolution (500–100 nm/px) map of individual sections. Cells were identified at low resolution, and at least 30 cells were chosen at random and captured at high resolution (4–8 nm/px) for each condition.

### Agilent Seahorse Mito Stress Test

iMGs were plated onto Cell-Tak-coated 96-well Seahorse Agilent plate 1 day before the assay. The day after, microglia media was removed and cells were incubated in 180 µl of solution containing phenol-red-free DMEM media enriched with 1 M Glucose solution, 100 mM Pyruvate Solution and 200 mM Glutamine solution. The Agilent Mitochondrial Stress Test kit was applied per the manufacturer's instructions.

### In vitro image analysis

Integrated density (measured per cell) for fixed and live iMG and co-culture fluorescent staining was measured using ImageJ software.

### Neural progenitor cells (NPCs)

iPSCs were plated at low density to allow growth of individual colonies. Once colonies reach around 50–100 cells in size, they were lifted from the plate and transferred onto 6-well ultra-low attachment plates

(Corning, #CLS3471) placed on a shaker for embryoid body formation. On Day 17, embryoid bodies were plated onto PLO/Laminin-coated 10 cm plates for rosette formation. On Day 24, the most developed rosettes were manually selected and enzymatically dissociated using Accutase (StemCell, #07920). The resulting cell solution was plated onto PLO/Laminin-coated plates for neural progenitor cell expansion. NPCs were maintained in NPC media, containing DMEM/F12 with 1X Glutamax (ThermoFisher Scientific), 1X B27 Supplement (Gibco), 1X N-2 Supplement (Gibco), 1X Laminin (Invitrogen, #23017-015) and 1X FGF2 (Joint Protein Central, Korea).

### Neuron differentiation and co-culture

NPCs were plated in glass-bottom 8-well slides (Millipore Sigma, #PEZGS0816) at a density of 60k/0.7 cm² and cultured in NPC media for 24 h. Media was then exchanged for neuron differentiation media consisting of DMEM/F12 with Glutamax, 1% B27 Supplement (Gibco), 1% N2 Supplement (Gibco), 20 ng/mL BDNF (Peprotech, #450-02-1 mg), 20 ng/mL GDNF (Peprotech, #450-10-1 mg), 0.5 mg/mL dibutryl cyclicAMP (cAMP) (Tocris Bioscience, #1141) and 0.2 µM ascorbic acid (StemCell Technologies Cat. #72132) for 7 days. Mature iMG were resuspended in co-culture media, which consisted of neuron differentiation media supplemented with 25 ng/ml M-CSF (Humanzyme, #HZ-1039), 100 ng/ml IL-34 (Humanzyme, #HZ-7064), and 50 ng/ml TGFb-1 (Humanzyme, #HZ-1087). Microglia were then added to neurons at a ratio of 1:5 and co-cultured for 2 days. Co-culture was fixed, stained and imaged usingan immunofluorescence protocol. For co-culture live imaging, neurons were labeled with 1:1000 CellTrace CFSE for 20 min at 37 °C and microglia were labeled with 1:1000 CellTrace Far Red for 20 min at 37 °C and then added to neurons at a ratio of 1:5. After 2 days of culture, FerroOrange was utilized to stain for iron accumulation according to manufacturer's instructions.

### In differentiation and co-culture

iPSCs were transduced with lentiviral particles pLVX-UbC-rtTA-Ngn2:2 A:EGFP (UNG) and expanded in the presence of 1 µg/ml puromycin (Gibco, #A1113803) for 24 h then 0.5 µg/ml puromycin for an additional 24 h. UNG-iPSCs were then plated in monolayer and 24 h after plating, were induced with 2 µg/ml doxycycline (Stemcell, #72742) for 48 h. UNG-iPSCs were switched to Neuron Maturation Medium (NMM) supplemented with 2 µg/ml doxycycline and 100 µM AraC for 24 h. Neuron Maturation Medium consisted of DMEM/F12 and Neurobasal A (Gibco) (1:1), 1% N2 Supplement, 1% B27 supplement, 2 µg/ml Laminin (Invitrogen, #23017-015), 10 ng/mL BDNF (Peprotech #450-02-1 mg), 10 ng/mL GDNF (Peprotech #450-10-1 mg), 0.5 mM dibutryl cyclicAMP (cAMP) (Tocris Bioscience #1141) and 5 ng/mL NT-3 (Stemcell, #78074). iNs were then plated in 8-well slides with 200k cells/well in NMM supplemented with 2 µg/ml doxycycline, 20 µM AraC and 1 µM Rock inhibitor (Fisher Scientific, #HY-10583-10MG). Medium was changed the next day to NMM with 20 µM AraC, and a half media change with NMM was completed every other day. After 7 days, iMG were collected and labeled with CellTrace CFSE for 20 min at 37 °C and added to neurons at a ratio of 1:10. After 2 days of co-culture, FerroOrange was utilized to stain for iron accumulation according to manufacturer's instructions.

### Conditioned media

To generate conditioned media iMG were plated at a density of 150k cells/well in a 24-well plate. The media was collected 48 h later and was centrifuged at 300 g for 5 min to remove all cells. The supernatant was collected and diluted with neuron differentiation media at a ratio of 1:1 and was added to neurons for 48 h.

### Mice

Mice utilized in the present study were maintained at the Sanford Consortium for Regenerative Medicine. Humanized immunodeficient mice were purchased from the Jackson Laboratory (Strain #017708) (CSF1h/h) and subsequently depleted of murine microglia (CSF1RΔFIRE/ΔFIRE), generated by Drs. Clare Pridans and David Hume[56]. Mice were maintained in accordance with the UCSD research guidelines for the care and use of laboratory animals under a 12 h light/12 h dark cycle at 22 °C. We have complied with all relevant ethical regulations. Animal protocol # S19162. All animal procedures were approved by the Institutional Animal Care and Use Committee of California (IACUC).

### Intracerebroventricular engraftment (ICV) of iHPCs

iHPCs from the 264 healthy donor cell line, 223 FRDA and 223 edited were collected as previously described, and a density of 100k was resuspended in 4 µl of 1x PBS per pup. Intracerebroventricular injection (i.c.v) was performed in neonatal mice (P0-P2) by bilateral injection of 2 µl of PBS containing 50k of iHPCs per side per mice targeting the lateral ventricle, as previously described[6].

### Tissue processing

Mice were matured to adulthood (approximately 2 months/60 days), then were deeply anesthetized and intracardially perfused with 0.1 M PBS (pH 7.4) followed by 4% paraformaldehyde (PFA) solution. The brain was post-fixed with 4% PFA solution overnight at 4 °C and subsequently cryoprotected in 30% sucrose solution (in PBS) for 48 h at 4 °C prior to embedding in Tissue-Tek® O.C.T. Tissue was then sliced at 40 µm thick sections and freefloating sagittal cerebellar slices were cut using a cryostat and collected in PBS + 0.8% sodiumazide (AxonLab) at 4 °C.

### Immunohistochemistry of mice cerebellar sections

Cerebellar sections were blocked using 10% NDS + 0.5% Triton-X100 for 2 h at RT, followed by overnight incubation with primary antibody in 3% BSA + 0.5% Triton X100 in PBS at 4 °C. Following incubation with primary antibody, the sections were washed thrice with PBS and incubated for 2 h at room temperature with appropriate secondary antibodies. The sections were then rinsed again with PBS, stained with DAPI and subsequently mounted on glass slides using Immu-mount.

### Immunohistochemistry of human cerebellar sections

Human cerebellar sections underwent heat-induced epitope retrieval (HIER) with IHC antigen retrieval solution at 50 °C for 20 min. Tissue sections were then allowed to cool to room temperature before being washed with PBS twice. Sections were blocked using 10% NDS + 0.5% Triton-X100 for 1.5 h at room temperature and subsequently incubated overnight with primary antibodies in 3% BSA + 0.5% Triton X-100 in PBS at 4 °C. Following primary incubation, sections were washed thrice with PBS and incubated for 2 h at room temperature with appropriate secondary antibodies. The sections were then rinsed again with PBS, stained with DAPI and subsequently mounted on glass slides using Immu-mount.

### Confocal microscopy

Confocal images were acquired from lobule V-X using an Andor Dragonfly spinning confocal microscope (Oxford Instruments), fitted with 10×, 20×, 40× or 63× objective. Identical confocal settings were maintained during acquisition, and all the images were processed and acquired simultaneously.

### In vivo image analysis

Microglia migration and density were performed using ImageJ software by dividing each lobule into designated regions of interest (ROIs) and manually counting the number of Ku80+ cells populating each layer of the cerebellar cortex. Microglia density from WT, granule cell layer (GCL), Purkinje cell layer (PCL) and molecular cell layer (MCL) were assessed and plotted using GraphPad Prism. Microglia morphology was investigated using the filament tracing module in Imaris

software (Oxford Instruments). Results obtained from the microglial sholl analysis were plotted as number of intersections per radius. CD68[+] and 8-OHdG[+] puncta were imaged through cerebellar layers using 40x glycerol objective with 0.5 μm step size. Imaris software was then used to create a 3D view of the z-stacks and to build the 3D surface for each CD68[+] and 8-OHdG[+] puncta in IBA1 and/or Calbindin[+] cells present on the visual field of randomly designed ROIs of similar area. Puncta were identified if present in at least two consecutive optical sections, with a puncta diameter threshold set at 0.4 μm. To quantify the density of puncta expressed by cells, only 3D surfaces with a distance of 0 μm between the two 3D surfaces were considered for the analysis and plotted using GraphPad Prism. To quantify PC number, only cerebellar lobules with a clear Calbindin and IP3R staining were included in the analysis in all the conditions.

### Statistical analysis

Statistics were performed using GraphPad Prism 10 software (Graph-Pad Software Inc.). For all experiments, values from at least three biological replicates were used and presented as a mean ± standard error of the mean (SEM), as a bar graph or as a violin plot. One-way or Two-way ANOVA followed by post hoc Bonferroni's test was used to compare multiple groups. Statistical significance was defined as $*p < 0.05$; $**p < 0.01$; $***p < 0.001$. Statistical details of the experiments are described in the respective figure legends.

### Reporting summary

Further information on research design is available in the Nature Portfolio Reporting Summary linked to this article.

### Data availability

All data supporting the findings of this study are included in the article and its supplementary materials. Source data are provided with this paper. Any additional requests for information can be directed to the corresponding author. Source data are provided with this paper.

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

## Acknowledgements

These studies received support from the Friedreich's Ataxia Research Alliance (FARA) (N.G.C. and S.C.), the National Institutes of Health (NIH) R01NS135162 (N.G.C. and S.C.), and the California Institute for Regenerative Medicine (CIRM TRAN1-13983; S.C.). A.J., S.G., C.M., and E.M. were supported by CIRM grant EDUC2-08388. C.P. was supported by a fellowship from FARA. This publication includes data generated at the UC San Diego Human Embryonic Stem Cell Core Facility utilizing an Incucyte S3 that was purchased with funding from a National Institutes of Health grant (#1S10ODO25060-01). Figure schematics were made with Biorender.

## Author contributions

Conceptualization, C.P., A.J., S.G., S.C., and N.G.C.; electron microscopy, S.W-N., cell line establishment, A.J., E.H.B., C.P., A.S., and P.M.; Tissue culture experiments, A.J., C.P., S.G., E.H.; in vivo experiments, C.P., G.R.; Post-mortem human experiment, C.P. and C.M.; Human transcriptomic dataset, A.W.-S. Resource acquisition N.G.C. and S.C.

## Competing interests

S.C. is a cofounder, shareholder, and member of both the Scientific Board and board of directors of Papillon Therapeutics Inc. S.C. also serves as a member of the Scientific Review Board and Board of Trustees of the Cystinosis Research Foundation. This work is covered in the patent entitled "Methods for treating Mitochondrial Disorders" #114198-3029; (inventor: S.C.; granted U.S. Patent Nos. 12,012,436, 12,012,437 and 12,011,488, issued on June 18, 2024, covering the use of the gene-edited HSPCs for Friedreich's ataxia and the specific gene editing approach employed. A.S. is a paid consultant for Papillon Therapeutics Inc. The

terms of this arrangement have been reviewed and approved by the University of California, San Diego in accordance with its conflict-of-interest policies. All other authors declare no conflict of interest.
