## [Transparent Peer Review file · Nature Communications]

Microgliopathy as a primary mediator of neuronal death in models of Friedreich's Ataxia.

Corresponding Author: Dr Nicole Coufal

Version 0:

Reviewer comments:

Reviewer #1

(Remarks to the Author)

In this manuscript, Pernaci and colleagues investigate the properties of FRDA microglia and explore their potential cell-autonomous role in neuronal toxicity. In the first part of the study, the authors report that FRDA patient-derived microglia, differentiated from iPSCs, exhibit reactive morphology, mitochondrial dysfunction, reduced oxidative metabolism, increased phagocytosis, impaired lysosomal function, increased Fe²⁺ and ROS levels, as well as altered lipid metabolism and DNA damage. In co-culture experiments, these patient-derived microglia induce DNA damage and apoptosis in iPSC-derived healthy neurons. Notably, all these effects are reversed in FRDA microglia where the FXN expansion has been previously corrected via gene editing (isogenic lines). In the latter part of the study, the authors show that ICV injection of HSCs derived from FRDA-patient iPSCs into microglia-depleted mouse pups results in cerebellar infiltration, differentiation into reactive microglia, and PCs death. This effect appears to be specific to CAG-expanded FXN, as isogenic CRISPR-Cas9-corrected microglia do not impact PCs viability.

The topic of this manuscript is highly relevant, as the cellular mechanisms contributing to neuronal degeneration in FRDA remain relatively underexplored. In particular, the role of microglia in disease onset and progression has been studied to a lesser extent compared to other neurodegenerative disorders such as AD or ALS, where the non-cell-autonomous impairment of vulnerable neuronal populations was established years ago. That said, while this study provides valuable insights into FRDA microglial dysfunction, the observation that FRDA microglia exhibit alterations is not entirely novel. Previous studies, that should be properly cited, have shown that frataxin deficiency in microglia leads to increased DNA damage, a reactive phenotype characterized by pro-inflammatory factor production, impaired migration and phagocytosis, mitochondrial dysfunction with reduced respiration, a metabolic shift toward glycolysis, and elevated ROS levels (Shen et al., Plos ONE 2016; Della Valle et al., Genes Dis 2023; Sciarretta et al., 2024). In addition, the neurotoxic effect of FRDA microglia on neurons has been already demonstrated in culture (Della Valle et al., Genes Dis 2023).

The key strength of this study lies in the fact that its findings, which align with previous literature, are derived from microglia generated from FRDA patient iPSCs. This approach provides a valuable human-based model to investigate disease mechanisms. Additionally, regarding microglial phenotype characterization, this study presents novel data on autophagy, iron content, and lipid dysmetabolism, features that have not been extensively characterized in FRDA microglia. Furthermore, the investigation of HSCs transplantation builds upon previous studies from the same group (Rocca et al., Sci Transl Med, 2017; Rocca et al., Mol Ther Meth & Clin Dev, 2020) and adds an important piece of evidence to FRDA pathogenesis. Specifically, it highlights that mutant FXN in microglia is sufficient to induce PCs death in vivo, supporting the notion of a microglia cell-autonomous mechanism contributing to the disease.

For all these aspects I find this manuscript very interesting and informative about the role of microglia in FRDA. However, despite my overall positive assessment, I have some concerns regarding the scientific soundness of certain experiments, as well as the way some data are presented. Below, I outline several major issues that should be addressed by the authors:

1) The manuscript lacks clarity on the number and selection of cell lines used across different experiments. For instance, FXN expression characterization (a crucial control for cell line validity) includes 2 controls, 3 FRDA, 2 gene-edited, and 2 familial carrier lines. However, other experiments, such as ramification count analysis, use 4 FRDA and 3 isogenic gene-edited lines. Why is the 850 FRDA cell line and its isogenic control, which the authors claim to have generated, not consistently included? Additionally, in the FXN expression WB graph, the legend states, "Each dot represents a biological replicate with N=3 independent experiments/line." How many technical replicates were used per experiment, and why do their numbers differ across conditions? Furthermore, the FXN blot should be shown as a complete image, with all lines in separate lanes (11 total), which can be accommodated within a single gel.

2) What percentage of differentiated cells are mature microglia? Are macrophages also present? Iba1 and CD68 do not

distinguish microglia from macrophages, so a microglia-specific marker (e.g., TMEM119) is required. FXN staining should be counterstained with such a marker to ensure specificity.

3) It is unexpected that CD68 marks all cultured cells, both FRDA and gene-edited, and that it is expressed in both amoeboid and elongated cells. Similarly, the increase in Iba1 expression (that by the way seems the opposite in the image) in FRDA microglia is surprising, as Iba1 is not typically considered a reactivity marker *in vitro*. The authors should clarify the rationale behind these findings and provide a separate Iba1 channel for better visualization. Moreover, the Authors should include images from all experimental groups, particularly for Figure 1f, where the carrier group appears to exhibit a similar trend in FRDA microglia.

4) Why was mitochondrial impairment induced with CCCP to study autophagy when FRDA microglia already exhibit dysfunctional and fragmented mitochondria? The necessity of this intervention should be justified.

5) FRDA microglia show increased phagocytic activity compared to gene-edited cells. However, phagocytosis requires ATP, and FRDA microglia have reduced ATP production. How do the authors reconcile this apparent contradiction? Moreover, lipid droplet accumulation is often associated with reduced phagocytosis in impaired microglia (e.g., Marschallinger et al., *Nat Neurosci*, 2020). The authors should address these inconsistencies. Additionally, the Authors should provide representative images for the phagocytosis experiments (Figure 3c).

6) The γ H2AX signal in FRDA cells (Figure 4e) appears oversaturated. The authors should ensure proper image exposure.

7) The neuronal co-culture includes astrocytes. How is the composition of neuronal and glial populations controlled to prevent variability in astrocyte contributions across experiments?

8) In Figure 5c, neurons should be co-stained with β -tubulin to confirm that the ferro-orange signal is neuron-specific.

9) Unlike in Figure 4e, Figure 5d does not show huge γ H2AX staining (as shown in FRDA microglia). If microglia induce neuronal DNA damage via cell-interaction mechanism, one would expect reactive microglia surrounding damaged neurons. This discrepancy should be clarified.

10) The caspase-3 staining is difficult to interpret. Where is the signal localized? Higher-magnification images would help clarify its distribution.

11) The lack of effect from FRDA microglia-conditioned media on neuronal viability appears contradictory to the increased ROS levels observed in FRDA cells. This suggests a pro-oxidant intra- and extracellular environment. The authors should examine whether NOX2, the main extracellular ROS-producing enzyme in microglia, exhibits differential activity across cell lines. Moreover, they should analyze ferroptosis, mitochondrial damage, and changes in neuronal morphology after conditioned medium challenge across the different experimental groups.

12) The authors should discuss the rationale behind the *in vivo* experimental design, specifically why neonatal mice were injected and why 8 weeks was chosen as the endpoint.

13) The manuscript does not specify which cerebellar lobule was analyzed. Consistency in lobule selection is critical for reliable results. The authors should clarify this point.

14) The *in vivo* activation state of microglia should be further characterized given its strong effects on neuronal survival. The authors should analyze ROS levels as well as iNOS and NOX2 expression.

15) In Figure 6, the authors should include untreated control mice, particularly for calbindin staining. Moreover, in Figure 6f, the condition of other neuronal populations, such as cerebellar granule cells, should also be shown.

16) Figure 6b shows fewer cells in the GCL in the FRDA condition compared to the edited group. The authors should explain this discrepancy in relation to the graphs. Additionally, in Figure 6d, they should provide a larger image representing CD68 staining across all cerebellar layers.

17) The loss of P2Y12 signal is not clearly visible in the provided images. Higher-quality images and direct quantification would strengthen this observation.

18) In the *in vivo* experiments, HSCs were injected without sorting. Differentiation from iPSCs to HSCs can yield varying percentages of CD34+ cells across lines. Typically, HSCs are purified before injection to ensure consistency. The authors should clarify whether unsorted cells were injected and discuss how this may impact data accuracy.

19) The Discussion primarily reiterates the Results. The authors should better contextualize their findings within the existing literature on FRDA microglia.

20) The study reports FRDA microglia toxicity to PCs. However, in both human FRDA and mouse models, PCs are largely spared, while dentate nucleus neurons are the primary degenerating population. The authors should discuss why FRDA microglia target PCs in their model, considering the known selective vulnerability in FRDA.

Minor points:

1) Figure Supplementary 2a: How do the authors explain the decrease in non-mitochondrial OCR in FRDA cells, given that this parameter is typically expected to increase, for instance, due to NOX2 activity in conditions of mitochondrial damage and enhanced phagocytosis?

2) The Materials and Methods section contains several typos. Notably, in the ICV injection paragraph, the volume of cell suspension should be 4 μ L, not ml.

Reviewer #2

(Remarks to the Author)

Reviewer #3

(Remarks to the Author)

The involvement of microglia in FRDA pathogenesis has been established in a few published studies, as mentioned by the authors. However, the main impactful message of this paper is the authors' claim that microglia may be a primary driver mediating neuronal death in FRDA independent of the FXN mutation in neurons. This claim is based on in vitro co-culture studies using FRDA iPSC-derived microglia with healthy iPSC-derived neurons as well as in vivo xenotransplantation studies. Whilst the findings described in this manuscript are of high significance to FRDA research, the overall presentation of the existing data and substantial further studies are both needed to support the authors conclusions.

One of my main concerns is that most of the data showing prominent disease-related phenotypes in FRDA iPSC-derived microglia was combined from 3 FRDA iPSC lines and compared to combined results from 2 gene edited (or corrected) FRDA iPSC lines. I'm concerned that the additional FRDA iPSC line may have a more severe phenotype than the other two FRDA iPSC lines and thus skew the overall result to show a greater difference between diseased and corrected lines. Is there a significant difference in the data when comparing samples from individual FRDA iPSC and its corresponding gene edited line? I appreciate that the data was also compared to FRDA carrier and healthy control samples, but given the variability of iPSC lines in general, analyses across many lines need to be performed when they are not from the same genetic background (especially for publication in Nat Comm).

As mentioned above, the impactful message of the paper is that healthy neurons are impacted by FRDA microglia and thus may be a primary driver of neuronal death in FRDA. However, minimal data was provided to validate this claim. The in vitro co-culture studies are using relatively immature neurons (differentiated for one week), which may be more sensitive to hyperactivated microglia. Furthermore, the increase in neuronal iron levels may lead to ferroptosis, which was not looked at. Most of the in vivo data was validating the microglial phenotype in vivo and very little data on neurons was presented – only expression of Calbindin staining. There was no other neuronal marker/s used to determine whether loss of neurons was more widespread in this region. This needs to be done, particularly given the main claim of this manuscript. It would also be expected that the mice show a behavioural phenotype with Purkinje neuronal loss. Although it's mentioned in the discussion that this would be part of future work, behavioural studies would be expected for this high calibre journal.

The increased migration of FRDA microglia in the Purkinje layer is interesting. Was the same phenotype observed in the human FRDA post-mortem tissue shown in Figure 1a? How does this finding compare to other published FRDA human or mouse studies?

If the microglia are driving neuronal death, as suggested in the title, then a more widespread phenotype may be expected in the brain of a FRDA individual. Can the authors please comment on this.

Other specific comments:

1. In all of the figures (except 1e) representative images are provided for the FRDA and edited cells but not the control or carriers. These are also not present in the supplementary. This should be included.
2. The conclusion of figure 1 is that the FRDA microglia exhibit an activated phenotype and morphology. To confirm this, a positive control of activated microglia is needed (e.g., inducing an activated state in the cell lines) to validate that the observed differences in the FRDA microglia are truly indicative of microglial activation.
3. Fig.1e authors note "revealed appropriate repopulation of Tom20+ mitochondria with FXN edited lines compared to their respective isogenic FRDA lines" was this result quantified? Was there a loss of Tom20 expression (i.e., reduction in mitochondria) in the FRDA lines?
4. Fig.2d: There is a significant reduction in mitochondrial membrane potential (decrease in mitotracker red), is this normalised to the total number of mitochondria present? If there is a reduction in Tom20 expression (fig. 1e) then the reduced MTDR integrated intensity could be due to the reduction in total mitochondria rather than membrane potential. Please clarify.
5. However, in fig.2f it is reported that there is an increase in the total mitochondria. Please clarify.
6. To investigate the microglia morphology (Fig. 1f) it would be beneficial for the analysis method to have more detail. For example, a schematic or example image explaining how the software detected ramification branch points and length is required. Other parameters of morphology such as circularity, shape factor and eccentricity could also be beneficial in the analysis.
7. In Figure 5, it is said that the cultures are comprised of neurons and astrocytes, while there are stains of β -Tubulin it would be helpful to include stains of astrocyte and glial markers. Additionally, it would be good to have a quantification of the proportions of astrocytes/glia:neurons in the cultures.
8. In Figure 5, control samples of neuron/astrocyte alone (no added microglia) is required for all conditions as it is unclear what the baseline FerroOrange, H2AX, Caspase 3 and blebbing is in these cultures and whether the addition of any microglia improves or impairs the neuron/astrocyte cultures.
9. Fig.5c it is noted in the figure legend that the neuron/astrocyte cultures were labelled with CellTrace prior to microglia addition. It would be helpful to include representative images of the CellTrace + FerroOrange to identify the difference between the iMG and neurons/astrocytes. Otherwise, it is unclear which cells contain the FerroOrange. Additionally, based on the representative images the presence of FerroOrange appears to be similar between FRDA and edited iMG conditions.
10. Similarly, an overlay of the CellTrace in fig.5e is also needed. Based on the representative images the β -Tubulin does not look like it overlays with the Caspase-3, could the +Caspase-3 be present in the nonneuronal component of these cultures?
11. Fig.5f: Is there a difference in total β -tubulin staining?
12. Additional cell death assays are required to determine whether the addition of microglia causes neuronal cell death. An important measure of neuronal health is the functionality of the neurons. Microglia also have known roles in altering neuronal excitability. It would strengthen this work if the addition of microglia alters/impairs the functionality/excitability of the neuron/astrocyte cultures.

13. Fig.6b: Based on the representative figure, it looks like there is more Ku80 (human nuclei) present in FRDA condition. Is this related to the transplant efficiency or proliferation of the transplanted cells? Could the effects seen in the later panels (c-f) be due to more microglia being present in the FRDA transplanted mice?

14. Fig. 6f: The loss of neurons following microglial transplantation is quantified through a decrease in calbindin staining. However, to make the claim that microglia drives neuronal loss more quantification methods are required.

Minor:

Fig. 2d Spelling error "Integrated"

Fig.2a/b: Is the MitoSOX integrated density normalised to the mitotracker green?

Spelling/formatting errors within the methods sections.

Reviewer #4

(Remarks to the Author)

Reviewer #5

(Remarks to the Author)

The manuscript by Pernaci et al explores the impact of FRDA pathogenic expansions in the biology of microglia. This has been partially explored before in Frataxin deficiency mouse and cellular models (PMID: 38631900), but not in iPSC derived microglia and with this level of detail. The authors provide a robust and convincing in vitro characterization across several microglial functions and subcellular compartments, and performed a small proof of concept experiment in vivo using a xenotransplantation model. Although I find the data and premises of this work interesting, there are several aspects (both technical and conceptual) that diminish my enthusiasm.

1) The idea of CRISPR genetic editing as therapeutic strategy is interesting and some groups have had recent breakthroughs on this area. However the current manuscript comes really short at proving this is a valid strategy in humans, as it only shows that FRDA microglia is altered and can cause alterations in other cells. They do not provide any evidence that corrected microglia can reduce the deficits in FRDA neurons. Thus, although the concept is interesting, this idea is not supported by any of the data provided here. If the authors intend to propose the editing and transplantation of microglia into patients, it is not supported here that this would have any impact on pathology (the authors should take a FRDA mouse model and transplant it with corrected microglia. Alternatively, if the authors suggest using CRISPR editing in vivo, they should provide at least proof of concept data that this strategy can work in vivo in their xenotransplantation model.

2) The key experiment on this manuscript is the in vivo part, but it is very underdeveloped. Besides the CD68 quantification, are there any other indications that the several observations from the in vitro experiments are valid in vivo? Do microglia show alterations in phagocytosis, lipid peroxidation, etc? And do neurons display DNA damage or increased Caspase-3 expression?

3) The entire manuscript is merely descriptive, and does not provide any mechanistic insights as to how 1) the FRDA mutation leads to microglial cell autonomous functional alterations, and most importantly 2) by which mechanisms the altered microglia can modify neuronal physiology. I feel this is a missed opportunity and more mechanistic experiments should be provided.

4) The in vitro part is generally robust and well performed. However, I noticed a number of aspects that need to be improved

a) In figure 1. It is not clear how many cells were quantified per human sample in figure 1a and b. In panels f and g, there are clear differences in the density of cells based on the images provided. Cell density has a strong impact on iPSC microglia, both at morphological and phenotypic level. How do the authors rule out that there is no such impact in their system? In this same line the quantification of CD68 and Iba1 in Figure 1g does not seem to be normalized per cell, therefore also introducing the confounding element of cell density. There are also discrepancies between the images and quantifications. From the images, the edited microglia seem to have as much (if not more) Iba1 signal than the FRDA cells; however the quantification does not show that. Why are the pictures from the other experimental not shown?

b) In figure 2. Again, the quantification of staining in vitro does not seem to be corrected by cell density, which may have a strong impact on the results. Why only pictures from only two experimental groups are displayed. The authors intend to claim that the Cas9 mediated excision of the pathogenic expansion normalized a series of readouts, but the representative examples of control cells are systematically omitted. Figure 2g and i appeared to be quantified on the basis of ROIs as individual datapoints. This is considered pseudoreplication, and statistics should be performed at the level of independent replicates.

c) In figure 3. In panel a it is shown that the mitophagy is actually not changed between the FRDA and the edited cells. The

link between mithophagy and phagocytosis is unclear. These are two fundamentally unrelated processes, that could be impaired in an independent way due to energetic and metabolic deficits induced by the pathogenic mutation. The quantification in figure 3f suffers from pseudoreplication.

d) The co-culture experiments are interesting and very provokative. However, to completely prove the impact of microglia in FRDA, additional experiments with FRDA neurons cultured with non-mutant microglia should be performed.

e) Across the different readouts shown here, the carrier line only partially mimics the FRDA group. What are the implications of this observation for the human disease?

5) The in vivo experiments are very interesting, but also very limited. It could greatly benefit from a larger sample size and more readouts. There are also some discrepancies that need to be addressed. For example, in figure 6 there seems to be inconsistencies between the images shown in panels b and f. Whereas in f it is shown PC loss, this is not clear in the images in panel b. Which one of the two observations is correct? The PC loss should be also be confirmed with additional stainings to rule out the findings are confounded by a downregulation of calbindin.

Version 1:

Reviewer comments:

Reviewer #1

(Remarks to the Author)

The authors have addressed the majority of the concerns raised in the previous round of review, particularly with regard to the in vitro experiments, which are now much more complete and convincing. While I still have some reservations about the in vivo data, I acknowledge the additional efforts made and believe the current version of the manuscript represents a significant improvement. I have no further major concerns.

Reviewer #2

(Remarks to the Author)

Reviewer #3

(Remarks to the Author)

The revised manuscript has substantially improved the quality of the presented data and supports the claimed conclusions.

Reviewer #4

(Remarks to the Author)

Reviewer #5

(Remarks to the Author)

Thanks to the authors for the revised version of their manuscript. The authors have addressed most of my comments regarding the in vitro part of the paper. However, there are still some critical points that need additional data. In the current form, I do not think the main message of this manuscript – microglia can be primary inducers of neuronal cell death in this form of Ataxia – is well supported. Below a list of aspects that can be improved.

1) Although I understand the limitations, the current manuscript still does not provide (or even speculate) on what potential mechanisms are involved in this putative physical induction of neuronal death via microglia. There are previous reports showing that during brain development, microglia can promote the degeneration of Purkinje cells in the absence of any, and mediated by physical interactions mutations (Marin-Teva et al. 2004). I suggest the authors do a thorough search in the literature.

2) The proposed main message of this manuscript is not well supported by the data. I think that the authors have provided a fantastic characterization of the microglia, and with no doubt one can claim that FRDA mutations primarily affect microglia. However, the claims of microglial mediated neuronal toxicity are not well supported. If the authors want to still claim that, they should provide additional data. Below I suggest a list of potential experiments.

a. Co-culture experiments using mutant iPSC-neurons and wild type microglia. This would address whether the impact of mutant microglia is relevant when neurons also carry a mutation. It could well be that the neuronal autonomous effect of the

- disease causing mutation will result in neurodegeneration regardless of the contribution of dysfunctional microglia.
- b. Is the cell death induction specific to Purkinje cells. I suggest performing a brain wide experiment looking at different anatomical regions to determine if this is a true selective vulnerability. I would presume that, if microglia are so dysfunctional, there would be other brain alterations.
- c. The in vivo experiment is very shallow and it is the key argument that supports the main claim of this manuscript. I have serious concerns that the observations are not the result of confounding factors. The authors used FIRE-KO mice for their experiments, which are a model of CSF1R-related leukodystrophy (REF). ALS typically presents with cerebellar ataxia. Alterations in cerebellar development and PC have been also observed in several mouse models (Kana et al. 2019; Marin-Teva et al, 2004; Nakayama et al. 2019, etc.). There is a substantial risk that the mouse model itself presents with cerebellar alterations, that can then be corrected with wild type but not FRDA microglia. Although this could be an interesting finding, it is not what the authors propose here. I suggest the authors check the cerebellar phenotype of their host mice in depth, and perform additional confirmatory experiments in mice that do not present a developmental lack of microglia. In fact, the only image provided in S7c show aberrant distribution, uneven size and morphological altered PCs in untransplanted vs. control mice.
- 3) "Notably, CRISPR/Cas9-mediated correction of the GAA repeat reverses microglial defects and mitigates neurodegeneration." This sentence in the abstract is misleading. The authors do not use any CRISPR based therapeutic approach, but simply correct and transplant non-mutant lines. Please adjust the abstract to fit to the data provided.
- 4) In figure 6b, the authors show an increase density of microglia in the areas where there is degeneration. Could it be simply that the dysregulated density of microglia is altering the physiology of the tissue?
- 5) There are still quantifications that suffer from pseudoreplication, such as 2h. Please correct this.

Version 2:

Reviewer comments:

Reviewer #5

(Remarks to the Author)

The authors have addressed my points satisfactorily. Thanks for the extra effort

Reviewer #1 (Remarks to the Author):

In this manuscript, Pernaci and colleagues investigate the properties of FRDA microglia and explore their potential cell-autonomous role in neuronal toxicity. In the first part of the study, the authors report that FRDA patient-derived microglia, differentiated from iPSCs, exhibit reactive morphology, mitochondrial dysfunction, reduced oxidative metabolism, increased phagocytosis, impaired lysosomal function, increased Fe²⁺ and ROS levels, as well as altered lipid metabolism and DNA damage. In co-culture experiments, these patient-derived microglia induce DNA damage and apoptosis in iPSC-derived healthy neurons. Notably, all these effects are reversed in FRDA microglia where the FXN expansion has been previously corrected via gene editing (isogenic lines). In the latter part of the study, the authors show that ICV injection of HSCs derived from FRDA-patient iPSCs into microglia-depleted mouse pups results in cerebellar infiltration, differentiation into reactive microglia, and PCs death. This effect appears to be specific to CAG-expanded FXN, as isogenic CRISPR-Cas9-corrected microglia do not impact PCs viability.

The topic of this manuscript is highly relevant, as the cellular mechanisms contributing to neuronal degeneration in FRDA remain relatively underexplored. In particular, the role of microglia in disease onset and progression has been studied to a lesser extent compared to other neurodegenerative disorders such as AD or ALS, where the non-cell-autonomous impairment of vulnerable neuronal populations was established years ago. That said, while this study provides valuable insights into FRDA microglial dysfunction, the observation that FRDA microglia exhibit alterations is not entirely novel. Previous studies, that should be properly cited, have shown that frataxin deficiency in microglia leads to increased DNA damage, a reactive phenotype characterized by pro-inflammatory factor production, impaired migration and phagocytosis, mitochondrial dysfunction with reduced respiration, a metabolic shift toward glycolysis, and elevated ROS levels (Shen et al., Plos ONE 2016; Della Valle et al., Genes Dis 2023; Sciarretta et al., 2024). In addition, the neurotoxic effect of FRDA microglia on neurons has been already demonstrated in culture (Della Valle et al., Genes Dis 2023).

We have cited these studies as requested.

The key strength of this study lies in the fact that its findings, which align with previous literature, are derived from microglia generated from FRDA patient iPSCs. This approach provides a valuable human-based model to investigate disease mechanisms. Additionally, regarding microglial phenotype characterization, this study presents novel data on autophagy, iron content, and lipid dysmetabolism, features that have not been extensively characterized in FRDA microglia. Furthermore, the investigation of HSCs transplantation builds upon previous studies from the same group (Rocca et al., Sci Transl Med, 2017; Rocca et al., Mol Ther Meth & Clin Dev, 2020) and adds an important piece of evidence to FRDA pathogenesis. Specifically, it highlights that mutant FXN in microglia is sufficient to induce PCs death in vivo, supporting the notion of a microglia cell-autonomous mechanism contributing to the disease.

We thank the reviewer for their positive comments.

For all these aspects I find this manuscript very interesting and informative about the role of microglia in FRDA. However, despite my overall positive assessment, I have some concerns regarding the scientific soundness of certain experiments, as well as the way some data are presented. Below, I outline several major issues that should be addressed by the authors:

1) The manuscript lacks clarity on the number and selection of cell lines used across different

experiments. For instance, FXN expression characterization (a crucial control for cell line validity) includes 2 controls, 3 FRDA, 2 gene-edited, and 2 familial carrier lines. However, other experiments, such as ramification count analysis, use 4 FRDA and 3 isogenic gene-edited lines. Why is the 850 FRDA cell line and its isogenic control, which the authors claim to have generated, not consistently included? Additionally, in the FXN expression WB graph, the legend states, "Each dot represents a biological replicate with N=3 independent experiments/line." How many technical replicates were used per experiment, and why do their numbers differ across conditions? Furthermore, the FXN blot should be shown as a complete image, with all lines in separate lanes (11 total), which can be accommodated within a single gel.

As is the case in most iPSC cohorts, certain lines were easier to differentiate than others, especially when it came to challenging microglial differentiations. The 850 FRDA cell line is the line with the highest number of GAA repeats (650/1030 repeats) perhaps contributing to challenges with the differentiation. Whereas neural progenitor differentiation was successful after multiple attempts (Mishra et al 2024, PMID 38420191), microglial differentiation, which requires continual return to iPSCs for ongoing differentiation, was substantially more difficult. Despite this limitation, we were able to successfully differentiate this line, and its isogenic gene edited pair and collected data for the following experiments now included in the manuscript Fig1.d and f; Suppl. 1 g-h, Suppl. 2 a-b; Suppl. 3f, Fig. 4d, Suppl. 7a. Additionally, a new FXN blot has been included in Fig1 d showing FXN levels from all the four FRDA patients' line and their respective gene edited controls.

2) What percentage of differentiated cells are mature microglia? Are macrophages also present? Iba1 and CD68 do not distinguish microglia from macrophages, so a microglia-specific marker (e.g., TMEM119) is required. FXN staining should be counterstained with such a marker to ensure specificity.

We thank the reviewer for this question and have addressed this in the revised manuscript. A new experiment has been added and displayed in Suppl. Fig 2A showing that 100% of differentiated cells are mature microglia exhibiting TMEM119 expression (both % of positive cells and by the comparative expression levels between different conditions). These data support our and other existing publications, showing that iPSC derived microglia are more similar to human microglia rather than macrophages (PMID 31375314, 30577865 and 28426964).

3) It is unexpected that CD68 marks all cultured cells, both FRDA and gene-edited, and that it is expressed in both amoeboid and elongated cells. Similarly, the increase in Iba1 expression (that by the way seems the opposite in the image) in FRDA microglia is surprising, as Iba1 is not typically considered a reactivity marker in vitro. The authors should clarify the rationale behind these findings and provide a separate Iba1 channel for better visualization.

To the best of our knowledge, CD68 can also be expressed by ramified microglia albeit at lower levels. Amoeboid and therefore activated microglia increase CD68 expression (PMID 32848611). We do agree with the reviewer that Iba1 expression levels per se are not sufficient to imply microglia hyperactivation if used in isolation as a marker for microglia inflammatory state in vitro. Elevation in Iba1 is widely regarded as occurring in inflammatory states (PMID 26048578) and also reported as increasing in vitro with inflammation (PMID 36359810) such as after LPS treatment (PMID 27075756). Here we used microglia morphology and CD68 expression levels, both reliable and well accepted assays to investigate microglia activation states in vitro and paired these with Iba1 expression.

To address the reviewer's concerns, we have now included a positive experimental control (See Suppl. Fig.2d) where we treated iMGs derived from two different healthy control lines with 100ng/mL of LPS for 24h, finding a significant increase of both CD68 and IBA1 expression levels over multiple independent biological replicates. Moreover, new representative images for CD68 and IBA1 expression levels have been added for all the conditions (See Fig. 1g FRDA and Edited and Suppl. Fg 2d for Control and Carrier).

Moreover, the Authors should include images from all experimental groups, particularly for Figure 1f, where the carrier group appears to exhibit a similar trend in FRDA microglia.

We thank the reviewer for this note. We have now included images from all experimental groups, adding controls and carriers in the supplemental given the large number of groups to prevent crowding of the primary figures.

4) Why was mitochondrial impairment induced with CCCP to study autophagy when FRDA microglia already exhibit dysfunctional and fragmented mitochondria? The necessity of this intervention should be justified.

We thank the reviewer for this comment. The dye we used (MD01-10; Dojindo) will specifically accumulate in healthy mitochondria, producing a weak fluorescence signal that will significantly increase at a pH4-5, therefore only after mitochondria fuse with the lysosomes. For this reason, we firstly incubated our cells with the Mitophagy dye for 30min to allow mitochondria labelling, we then induced mitochondria dysfunction to initiate the formation of autophagosome, using CCCP, that will terminate in phagosome-lysosome fusion resulting in fluorescence signal. At baseline conditions without CCCP, this dye failed to detect mitophagy in all conditions (See manufacturer instruction). We have also emphasized and clarified this point in the text.

5) FRDA microglia show increased phagocytic activity compared to gene-edited cells. However, phagocytosis requires ATP, and FRDA microglia have reduced ATP production. How do the authors reconcile this apparent contradiction? Moreover, lipid droplet accumulation is often associated with reduced phagocytosis in impaired microglia (e.g., Marschallinger et al., Nat Neurosci, 2020). The authors should address these inconsistencies. Additionally, the Authors should provide representative images for the phagocytosis experiments (Figure 3c).

We do agree with the reviewer that ATP production is a key contributor to the phagocytic capacity of microglia cells. However, ATP production via the OXPHOS pathway is not the only contributor that influences the phagocytic capacity of microglia. Indeed, a decrease in ATP production via OXPHOS can lead to a switch to the glycolysis pathway for ATP generation. Derangements in lipid metabolism in human iPSC microglia have also been linked to increases in phagocytic transcriptional pathways rather than reductions (PMID 38448474). Additionally, hyperphagocytic profile has also been observed in primary microglia derived from the FRDA KIKO mouse model where glycolytic function was upregulated compared to wild type microglia (Francesca Sciarretta et al., 2024 PMID 38631900). We have clarified this in the revised manuscript. Additionally, representative images have been added for all the conditions tested as requested (Fig. 3b and Suppl. Fig. 4b)

6) The γ H2AX signal in FRDA cells (Figure 4e) appears oversaturated. The authors should ensure proper image exposure.

We have provided a new image with better exposure in 4e as requested. Please note that the magnified image is a 3D surface reconstruction of γ H2AX puncta.

7) The neuronal co-culture includes astrocytes. How is the composition of neuronal and glial populations controlled to prevent variability in astrocyte contributions across experiments?

We thank the reviewer for this question and have clarified the text and methods. iPSC-derived neuronal progenitor cells (NPCs) were utilized to generate the neurons and astrocytes cocultures. All NPCs were differentiated from the same iPSC and plated for neuronal differentiation at the same passage and density to reduce variability in cell types and to prevent variability in astrocyte contribution. Additionally, astrocytes represent a small percentage of the co-culture as demonstrated in Supplementary Figure 6a.

8) In Figure 5c, neurons should be co-stained with β -tubulin to confirm that the ferro-orange signal is neuron-specific.

The FerroOrange dye is a live imaging dye that is not compatible with immunofluorescence. Neurons and microglia were labeled with cell trace prior to co-culture in order to differentiate microglia from other brain cell types. Neurons were also identified by morphology and nucleus size. To further address the reviewers' concerns, we performed FerroOrange staining in a new co-culture model composed of iPSC-derived direct induced neurons (iNs) generated through directed overexpression of the transcription factor NGN2, where the culture does not contain progenitors or astrocytes and then cocultured these with iMGs to confirm the phenotype of elevated FerroOrange in neurons after FRDA microglial coculture (see Supplementary Figure 6c).

9) Unlike in Figure 4e, Figure 5d does not show huge γ H2AX staining (as shown in FRDA microglia). If microglia induce neuronal DNA damage via cell-interaction mechanism, one would expect reactive microglia surrounding damaged neurons. This discrepancy should be clarified.

Representative images demonstrating microglia near γ H2AX positive neurons have been added to Figure 5g as requested. Unfortunately, we cannot compare γ H2AX staining between co-cultured healthy neurons and the quantitation of FRDA microglia cultured independently because they are different cell types, the experiments and immunofluorescence took place at different times. The numbers of FRDA iMGs in neuronal cocultures are small and not easily tractable for robust quantitation. Additionally, it is expected that FRDA microglia would exhibit more DNA damage as they contain the GAA repeat mutation while the healthy neurons are only cultured with microglia for 48 hours and DNA damage is an indirect consequence of dysfunctional FRDA microglia.

10) The caspase-3 staining is difficult to interpret. Where is the signal localized? Higher-magnification images would help clarify its distribution.

We have provided higher magnification images of caspase-3 in TUJ1 positive neurons in figure 5h as requested. The quantification, as noted in Figure 5i, is of caspase-3 specifically co-localized with TUJ1 positive neurons.

11) The lack of effect from FRDA microglia-conditioned media on neuronal viability appears contradictory to the increased ROS levels observed in FRDA cells. This suggests a pro-oxidant intra- and extracellular environment. The authors should examine whether NOX2, the main extracellular ROS-producing enzyme in microglia, exhibits differential activity across cell lines. Moreover, they should analyze ferroptosis, mitochondrial damage, and changes in neuronal morphology after conditioned medium challenge across the different experimental groups.

We agree with the reviewer that increased oxidative stress would suggest that perhaps conditioned media would be a significant contributor to neuronal death, thus sparking our interest in conditioned media experiments. However, the interactions between microglia and neurons are complex and multifaceted. To further support our results that neuronal death is primarily caused by cell-to-cell interactions, we completed further characterization of healthy neurons after treated with iMG conditioned media. Immunostaining for H2AX was completed as a measure of DNA damage in our neurons treated with iMG-CM (Suppl. Figure 6f-g), and did not show a difference whereas direct co-culture with iMGs caused elevated H2AX in neurons. We also quantified number of blebs, indicative of neuronal stress and apoptosis in our conditioned media model, again finding no difference whereas direct coculture increased neuronal blebbing (Suppl Figure 6i). These complement the existing data that there is not an increase in caspase 3 from iMG conditioned media, whereas direct coculture causes elevated caspase-3 in neurons. Altogether these results demonstrated no statistical significance between groups, which supports our previous conclusions. Of note, the conditioned media was generated from higher numbers of microglia per volume compared to direct microglia coculture, to try to elicit differences (see methods). We did query NOX2 as the reviewer requested, but the levels in conditioned media were below the level of detection.

12) The authors should discuss the rationale behind the *in vivo* experimental design, specifically why neonatal mice were injected and why 8 weeks was chosen as the endpoint.

Neonatal mice were used in this study to allow for a wide-spread repopulation of the empty microglia niche in the mouse brain allowing microglia progenitor cell to migrate and differentiate in fully mature xMG from development to adulthood. Previous studies have shown that hFIRE mice develop an adult onset leukodystrophy without microglia, which is fully corrected with neonatal transplantation (PMID 38897209). We chose the 8-week timepoint as an adult timepoint, where we have previously shown that xenotransplantation of iPSC derived hematopoietic progenitor's neonatally leads, at 8-weeks post transplantation, to microglia that transcriptionally and epigenetically closely recapitulate primary human microglia (Han et al *Immunity* 2023 PMID 37582369). We have clarified the text in this regard.

13) The manuscript does not specify which cerebellar lobule was analyzed. Consistency in lobule selection is critical for reliable results. The authors should clarify this point.

We thank the reviewer for this question and emphasized now in the text that we consistently focused on anterior cerebellar lobules (I-V) throughout the analysis.

14) The *in vivo* activation state of microglia should be further characterized given its strong effects on neuronal survival. The authors should analyze ROS levels as well as iNOS and NOX2 expression.

We thank the reviewer for this suggestion. We investigated iNOS expression *in vitro* and we did not find any statistically significant differences between genotypes (See suppl. Fig 3f). Similarly, we queried NOX2 protein levels in addition to the portion secreted that we couldn't detect in iMG media, and we did not find any statistically significant differences between genotypes. For this reason, we did not explore iNOS expression *in vivo*.

To address the reviewer's concern, we quantitated 8-Hydroxydeoxyguanosine (8-OHdG) levels, a well-accepted marker for oxidative stress used to detect oxidative damage of DNA as a consequence of ROS accumulation. We now show that FRDA xMGs exhibit an elevated oxidative phenotype consistent with our extensively characterized *in vitro* data (Fig. 2 and 4). Additionally, we now also showed that FRDA iMGs and xMGs trigger oxidative stress and DNA damage in healthy neurons both *in vitro* and *in vivo* (fig. 5d and g, fig. 6f and 7b).

15) In Figure 6, the authors should include untreated control mice, particularly for calbindin staining. Moreover, in Figure 6f, the condition of other neuronal populations, such as cerebellar granule cells, should also be shown.

We thank the reviewer for this suggestion. We have now included untransplanted controls (n=3) in our *in vivo* cohort (Fig. 7). We have quantified Purkinje cell number and granule neurons comparing untransplanted controls with mice xenotransplanted with iHPCs from healthy, FRDA patient and gene edited lines. These data are displayed in fig. 7a, c and Suppl. Fig 7c.

16) Figure 6b shows fewer cells in the GCL in the FRDA condition compared to the edited group. The authors should explain this discrepancy in relation to the graphs. Additionally, in Figure 6d, they should provide a larger image representing CD68 staining across all cerebellar layers.

We thank the reviewer for this comment and have selected a more representative image as requested.

17) The loss of P2Y12 signal is not clearly visible in the provided images. Higher-quality images and direct quantification would strengthen this observation.

We have included in the resubmission images that separate P2RY12 and HLADR channels as requested (Suppl. Fig 7b). We have clarified our discussion as the quantitation presented is specifically of the number of cells that are P2RY12 negative and HLA-DR positive, finding that this significantly increases with xenotransplantation of FRDA microglia (Fig. 6e and Suppl.7b)

18) In the in vivo experiments, HSCs were injected without sorting. Differentiation from iPSCs to HSCs can yield varying percentages of CD34+ cells across lines. Typically, HSCs are purified before injection to ensure consistency. The authors should clarify whether unsorted cells were injected and discuss how this may impact data accuracy.

We respectively disagree with the reviewer on this point. We (Hasselmann et al *Neuron* 2019 PMID 31375314, Han et al *Immunity* 2023 PMID37582369) and others (Chadarevian et al *JEM* 2023 PMID36584406, Mancusco et al *Nat Neurosci* 2019 PMID31659342, Chadarevian et al *Neuron* 2024 PMID 38897209 to name only a few) including the protocol papers on xenotransplantation (Fattorelli et al *Nat Protocols* 2021 PMID 33424025) have not performed additional purification such as flow cytometry prior to xenotransplantation. This has not been the standard in the field. However, to address the reviewer's concerns, we have queried the percentage of iHPCs expressing the early hematopoietic progenitor marker CD43 finding no difference between patient lines. (Suppl. Fig.7a).

19) The Discussion primarily reiterates the Results. The authors should better contextualize their findings within the existing literature on FRDA microglia.

We have revised the discussion as requested.

20) The study reports FRDA microglia toxicity to PCs. However, in both human FRDA and mouse models, PCs are largely spared, while dentate nucleus neurons are the primary degenerating population. The authors should discuss why FRDA microglia target PCs in their model, considering the known selective vulnerability in FRDA.

We do agree with the reviewer that DCN neuropathology is prominent in FRDA patients. However, Purkinje cells (PCs) are neuropathologically altered in patients with FRDA (Koeppen et al., 2016 *Plos pathogens* PMID: 27295279, Kemp et al., 2016 *Acta neuropathologica communications* PMID: 27215193) and PC loss has also been reported in an inducible model of FRDA pathology (Marcado-Ayon et al., 2022 *Frontiers in neuroscience* PMID: 35401081 and Joseph et al., 2025 *Cerebellum* PMID: 39907933, Vicente-Acosta et al, 2024 *Neurobiology of disease* PMID: 39111701).

1) Figure Supplementary 2a: How do the authors explain the decrease in non-mitochondrial OCR in FRDA cells, given that this parameter is typically expected to increase, for instance, due to NOX2 activity in conditions of mitochondrial damage and enhanced phagocytosis?

Non-mitochondria oxygen consumption measured by Seahorse assay gives an indication of oxygen consumed by oxygenase and enzymatic reactions that occurs outside and independently from mitochondria OCR. A decrease in non-mitochondria OCR indicates that our cells suffer from an overall decrease in metabolism. We did assay NOX2 protein levels finding no difference between patient lines.

2) The Materials and Methods section contains several typos. Notably, in the ICV injection paragraph, the volume of cell suspension should be 4 μ L, not ml.

We have corrected these inadvertent errors as requested.

Reviewer #3 (Remarks to the Author):

The involvement of microglia in FRDA pathogenesis has been established in a few published studies, as mentioned by the authors. However, the main impactful message of this paper is the authors' claim that microglia may be a primary driver mediating neuronal death in FRDA independent of the FXN mutation in neurons. This claim is based on in vitro co-culture studies using FRDA iPSC-derived microglia with healthy iPSC-derived neurons as well as in vivo xenotransplantation studies. Whilst the findings described in this manuscript are of high significance to FRDA research, the overall presentation of the existing data and substantial further studies are both needed to support the authors conclusions.

One of my main concerns is that most of the data showing prominent disease-related phenotypes in FRDA iPSC-derived microglia was combined from 3 FRDA iPSC lines and compared to combined results from 2 gene edited (or corrected) FRDA iPSC lines. I'm concerned that the additional FRDA iPSC line may have a more severe phenotype than the other two FRDA iPSC lines and thus skew the overall result to show a greater difference between diseased and corrected lines. Is there a significant difference in the data when comparing samples from individual FRDA iPSC and its corresponding gene edited line? I appreciate that the data was also compared to FRDA carrier and healthy control samples, but given the variability of iPSC lines in general, analyses across many lines need to be performed when they are not from the same genetic background (especially for publication in Nat Comm).

We appreciate the reviewer's concern. Most of our data arises from four different FRDA lines (Supplementary Fig. 1b) and three edited lines. We would like to point out that the fourth FRDA line for which no gene edited line was generated (FF1) is a full sibling of a second FRDA line (FF2) for which a gene edited line was generated. Please refer to Suppl. Fig. 1b for all the familial relationships between several lines. We therefore strongly feel that this is a robust cohort. As is the case in most iPSC cohorts, differentiation of some lines was easier than others, especially with difficult microglial differentiations. The 850 FRDA cell line is the line with the highest number of GAA repeats (650/1030 repeats) perhaps contributing to challenges with the differentiation. Whereas neural progenitor differentiation was successful after multiple attempts (Mishra et al 2024, PMID 38420191), microglial differentiation, which requires continual return to iPSCs for ongoing differentiation, was substantially more difficult. Despite this limitation, we were able to successfully differentiate this line, and its isogenic gene edited pair and collected data for the following experiments now included in the manuscript Fig1.d and f; Suppl. 1 g-h, Suppl. 2 a-b; Suppl. 3f, Fig. 4d, Suppl. 7a. Additionally, a new FXN blot has been included in Fig1 d showing FXN levels from all the four FRDA patients' line and their respective gene edited controls. We have therefore extended multiple experiments within the manuscript to include a third gene edited and a fourth FRDA line to address these concerns.

As mentioned above, the impactful message of the paper is that healthy neurons are impacted by FRDA microglia and thus may be a primary driver of neuronal death in FRDA. However, minimal data was provided to validate this claim. The in vitro co-culture studies are using relatively immature neurons (differentiated for one week), which may be more sensitive to hyperactivated microglia. Furthermore, the increase in neuronal iron levels may lead to ferroptosis, which was not looked at.

We thank the reviewer for this comment. We confirmed maturity of our neurons by staining for MAP2, a mature neuronal marker (Soltani et al 2005, PMID: 15920168), showing that NPC derived neurons expressed appropriate markers at the time points queried. Both in vitro and in

vivo microglia cells have been added at an early stage of neuronal development to best mimic their physiological role. Whether there are more pronounced differences in neuronal death depending on differentiation timepoints is not addressed in this study. Here we are showing that neurons expressing MAP2 shows marked cellular stress as reflected by neuronal blebbing, DNA damage, iron accumulation and upregulation of caspase-3 levels, readouts of an ongoing cell death program. Additionally, in this resubmission, we have extended on the data regarding neuronal cell stress and dysfunction caused by FRDA microglia and found that, in addition to iron accumulation, healthy mature neurons (both in vitro and in vivo) show a robust expression of 8-OHdG- a well-known marker for ROS induced neuronal damage (Fig. 5g, 6f and 7b). We agree that increase in iron levels as well as ROS accumulation can lead to ferroptosis which would require extensive further in vivo studies to definitively query the mechanism.

MAP2 Iba1 DAPI

Most of the in vivo data was validating the microglial phenotype in vivo and very little data on neurons was presented – only expression of Calbindin staining. There was no other neuronal marker/s used to determine whether loss of neurons was more widespread in this region. This needs to be done, particularly given the main claim of this manuscript. It would also be expected that the mice show a behavioral phenotype with Purkinje neuronal loss. Although it's mentioned in the discussion that this would be part of future work, behavioural studies would be expected for this high calibre journal.

We appreciate the reviewers point that additional neuronal markers would be ideal. To address this comment, we have now separated out Fig. 7 and included quantitation of neurons in the granule cell layer, as well as Purkinje neurons through two different orthogonal markers. Lastly, we have stained for the oxidative stress marker 8OHdG and have found increased Purkinje neuron oxidative stress in animals xenotransplanted with FRDA microglia (Fig. 7).

Regarding animal behavior, we respectfully disagree with the reviewer. The immunodeficient transgenic mice utilized for xenotransplantation are difficult to breed and to generate large cohorts for experiments such as behavior. In fact, to our knowledge *no group* has published behavior after xenotransplantation in this model. This includes in high caliber journals (Hasselmann et al *Neuron* 2019 PMID 31375314, McQuade et al *Nat Comm* 2020 PMID 33097708, Han et al *Immunity* 2023 PMID 37582369 among multiple others). Please note that we have now used IP3R as additional Purkinje cell marker, and we also quantified neuronal survival in the granule cell layer using NeuN. These data are displayed in fig. 7a, c and showed that neuronal death is not limited to Purkinje cells.

The increased migration of FRDA microglia in the Purkinje layer is interesting. Was the same phenotype observed in the human FRDA post-mortem tissue shown in Figure 1a? How does this finding compare to other published FRDA human or mouse studies?

We thank the reviewer for this comment. Unfortunately, due to limited access to human tissue samples and poor antibody efficiency, we were not able to obtain a Calbindin staining that allowed clear identification of cerebellar layers. We therefore focused on the molecular layer due to its easier recognition (based on the anatomical location). To the best of our knowledge, there are no studies that specifically dissect microglia distribution within white matter, granule cell layer, Purkinje cell layer and molecular cell layer. However, a recent publication from Andres Vincente-Acosta et al, 2024 *Neurobiology of disease* PMID: 39111701), investigated Iba1 intensity in the cerebellum but only in the white matter, molecular layer and granule cell layer with no information about the PC layer. They report an increase in Iba1 intensity in the YG8-800 mouse model as from three months of age specifically in the granule cell layer, therefore in close proximity to PC layer. Additionally, Iba1 intensity gradually increased by time terminating in overall Iba1 increase in the whole cerebellar cortex of 12months YG8-800 mouse model.

If the microglia are driving neuronal death, as suggested in the title, then a more widespread phenotype may be expected in the brain of a FRDA individual. Can the authors please comment on this.

FRDA patients and mouse models shows degeneration of mechanoreceptive and proprioceptive neurons in dorsal root ganglia, cerebellar dentate nucleus neurons and Purkinje cell loss (PMID 35401081, 14985441, 29257745). In FRDA postmortem tissue, degeneration of nerve fibers in the spinal cord leading to neuronal loss, degeneration of large neurons of the dorsal root ganglia, degeneration of dentate nucleus neurons and reduction in cerebellar peduncles have been reported, as well as patchy loss of Purkinje cells and inferior olivary nucleus neuronal degeneration (PMID: 19283344).

Other specific comments:

1. In all of the figures (except 1e) representative images are provided for the FRDA and edited cells but not the control or carriers. These are also not present in the supplementary. This should be included.

All representative images have now been included for almost all the experiments presented in the paper. However, due to limited figure space we selected to show only key pictures in the main figures (i.e. FRDA vs Edited), while control and carrier images will be found in suppl figures to prevent overcrowding of figures.

2. The conclusion of figure 1 is that the FRDA microglia exhibit an activated phenotype and morphology. To confirm this, a positive control of activated microglia is needed (e.g., inducing an activated state in the cell lines) to validate that the observed differences in the FRDA microglia are truly indicative of microglial activation.

We thank the reviewer for this suggestion. We have now included a positive experimental control (See Suppl. Fig.2d) where 100ng/mL of LPS treatment in two healthy donor line showed significant increase of both CD68 and IBA1 expression levels, indicating that increase in their expression can be used as proxy of microglia activation.

3. Fig.1e authors note “revealed appropriate repopulation of Tom20+ mitochondria with FXN edited lines compared to their respective isogenic FRDA lines” was this result quantified? Was there a loss of Tom20 expression (i.e., reduction in mitochondria) in the FRDA lines?

Virtually all FXN expression co-localized with TOM20, and we utilized TOM20 to show correct localization of FXN protein within mitochondria, without quantifying TOM20 via ICC-IF. We have now added a western blot showing that TOM20 protein levels (normalized over gapdh) are unchanged between different cell lines by genotype (Suppl. Fig 3e). We have clarified this statement in the text. Additionally, we extensively quantitated mitochondrial function in figure 2 using different approaches and we employ EM images to quantify mitochondria density and showed an accumulation of round and dysfunctional mitochondria in FRDA compared to respective controls.

4. Fig.2d: There is a significant reduction in mitochondrial membrane potential (decrease in mitotracker red), is this normalised to the total number of mitochondria present? If there is a reduction in Tom20 expression (fig. 1e) then the reduced MTDR integrated intensity could be due to the reduction in total mitochondria rather than membrane potential. Please clarify.

We are not reporting a reduction in TOM20+ mitochondria (Suppl. 3e), but rather an increase in mitochondria number assessed using EM images (Fig. 2g). The mitotracker deep red is normalized by cell given that live cell imaging is not detailed enough to clearly delineate mitochondrial number which requires EM.

5. However, in fig.2f it is reported that there is an increase in the total mitochondria. Please clarify.

We find that there is an increased number of fragmented mitochondrial in FRDA using EM and find that these fragmented mitochondrial are dysfunctional using multiple orthogonal measures including Mito tracker deep red and oxygen consumption. Given challenges with assessing precise organelle number by confocal microscopy, we elected for the more accurate EM for quantitation. We conclude there is accumulation of dysfunctional mitochondria.

6. To investigate the microglia morphology (Fig. 1f) it would be beneficial for the analysis method to have more detail. For example, a schematic or example image explaining how the software detected ramification branch points and length is required. Other parameters of morphology such as circularity, shape factor and eccentricity could also be beneficial in the analysis.

We have expanded on the details of the ramification analysis in the supplementary methods. This approach utilizes a commercially available live-cell imaging platform specifically quantifying ramification length and branch number rather than circularity or eccentricity, which are not measures captured from this software.

7. In Figure 5, it is said that the cultures are comprised of neurons and astrocytes, while there are stains of β -Tubulin it would be helpful to include stains of astrocyte and glial markers . Additionally, it would be good to have a quantification of the proportions of astrocytes/glia:neurons in the cultures.

We thank the reviewer for this comment and have now included Supplementary Figure 6a which has representative example of the different cell types present in the co-culture and a quantification of the proportions of the different cell types.

8. In Figure 5, control samples of neuron/astrocyte alone (no added microglia) is required for all conditions as it is unclear what the baseline FerroOrange, H2AX, Caspase 3 and blebbing is in these cultures and whether the addition of any microglia improves or impairs the neuron/astrocyte cultures.

All co-culture experiments do have a control sample that is neurons and astrocytes alone without microglia. This is represented by the purple bar in each graph and allows us to define the precise contribution of microglia is the investigated phenotype.

9. Fig.5c it is noted in the figure legend that the neuron/astrocyte cultures were labelled with CellTrace prior to microglia addition. It would be helpful to include representative images of the CellTrace + FerroOrange to identify the difference between the iMG and neurons/astrocytes. Otherwise, it is unclear which cells contain the FerroOrange. Additionally, based on the representative images the presence of FerroOrange appears to be similar between FRDA and edited iMG conditions.

Representative images that show neurons labeled with CellTrace CFSE dye have been added to Fig 5c to differentiate them from microglia in the co-culture. In addition, we have now also performed the FerroOrange with iPSC direct induced neurons, wherein neurons are induced through directed overexpression of the transcription factor NGN2 rather than through a neural progenitor intermediate, such that astrocytes are not generated. This also shows an increase in FerroOrange with FRDA microglial coculture (Supp Fig. 6c).

10. Similarly, an overlay of the CellTrace in fig.5e is also needed. Based on the representative images the β -Tubulin does not look like it overlays with the Caspase-3, could the +Caspase-3 be present in the nonneuronal component of these cultures?

We have provided higher magnification caspase-3 images in Figure 5h that demonstrate co-localization of caspase-3 and beta tubulin indicated by the arrows. Caspase-3 is a fixed antibody staining and therefore CellTrace is not utilized in this experiment, we apologize for any confusion.

11. Fig.5f: Is there a difference in total β -tubulin staining?

We did not specifically quantify the intensity of beta-tubulin, this is not to our knowledge a marker of neuronal degeneration or cell health.

12. Additional cell death assays are required to determine whether the addition of microglia causes neuronal cell death. An important measure of neuronal health is the functionality of the neurons. Microglia also have known roles in altering neuronal excitability. It would strengthen this work if the addition of microglia alters/impairs the functionality/excitability of the neuron/astrocyte cultures.

To further investigate the neuronal phenotype in our co-culture model, we stained for 8-OHdG, a marker for oxidative DNA damage and found that co-culturing with FRDA microglia resulted in an increase 8-OHdG-positive neurons. We further characterized neuronal death in our *in vivo* model as we found a significant reduction in Purkinje cells in the Purkinje cell layer of adult mice xenotransplanted with FRDA xMGs when compared to untransplanted animals and mice xenotransplanted with healthy and gene edited xMGs. While measuring neuronal functionality

and excitability would strengthen our understanding of the neuronal phenotype, this is outside the scope of the current study.

13. Fig.6b: Based on the representative figure, it looks like there is more Ku80 (human nuclei) present in FRDA condition. Is this related to the transplant efficiency or proliferation of the transplanted cells? Could the effects seen in the later panels (c-f) be due to more microglia being present in the FRDA transplanted mice?

The effect seen in panel c-f closely recapitulates what the phenotypes described through earlier figures, wherein microglia *in vitro* show an inflammatory phenotype and cause neuronal death when plated at the same density as the control conditions. An increased number of Iba1+ cells in the cerebellum recapitulates what is also reported in several recent publications Apolloni et al, 2022 PMID: 35682973 *Molecular Sciences*, Vincente-Acosta et al, 2024 *Neurobiology of disease* PMID: 39111701). We have not noted differences in transplant efficiency, with all brains, brainstem and spinal cord having full residence of myeloid cells throughout.

14. Fig. 6f: The loss of neurons following microglial transplantation is quantified through a decrease in calbindin staining. However, to make the claim that microglia drives neuronal loss more quantification methods are required.

We appreciate the comment and have undertaken additional experiments to address the reviewer's concern. We assessed neuronal loss by specifically quantitating the number of Calbindin+ neurons in 100µm of PCL therefore concluding that lower Calbindin+ neurons reflect neuronal loss. Additionally, we orthogonally validated by also staining Purkinje cells using IP3R and we again confirmed a similar reduction of IP3R+ neurons (Purkinje cells) in FRDA xenotransplanted mice. Furthermore, we also quantified the number of granule cell neurons using NeuN and we showed that neuronal death is not uniquely occurring in the Purkinje cell layer but rather a broad cerebellar neuronal degeneration might occur as a consequence of FRDA xMGs. These data are displayed in fig. 7a, c and Suppl. Fig 7c.

Minor:

Fig. 2d Spelling error "Integrated"

We have fixed this error, thank you.

Fig.2a/b: Is the MitoSOX integrated density normalised to the mitotracker green?

No, MitoTracker green was utilized to confirm MitoSox dye specificity, which has previously been shown to inappropriately stain the nucleus (PMID 26057935) in some protocols.

Spelling/formatting errors within the methods sections.

Resolved.

Reviewer #5 (Remarks to the Author):

The manuscript by Pernaci et al explores the impact of FRDA pathogenic expansions in the biology of microglia. This has been partially explored before in Frataxin deficiency mouse and cellular models (PMID: 38631900), but not in iPSC derived microglia and with this level of detail. The authors provide a robust and convincing *in vitro* characterization across several microglial

functions and subcellular compartments, and performed a small proof of concept experiment *in vivo* using a xenotransplantation model. Although I find the data and premises of this work interesting, there are several aspects (both technical and conceptual) that diminish my enthusiasm.

1) The idea of CRISPR genetic editing as therapeutic strategy is interesting and some groups have had recent breakthroughs on this area. However the current manuscript comes really short at proving this is a valid strategy in humans, as it only shows that FRDA microglia is altered and can cause alterations in other cells. They do not provide any evidence that corrected microglia can reduce the deficits in FRDA neurons. Thus, although the concept is interesting, this idea is not supported by any of the data provided here. If the authors intend to propose the editing and transplantation of microglia into patients, it is not supported here that this would have any impact on pathology (the authors should take a FRDA mouse model and transplant it with corrected microglia. Alternatively, if the authors suggest using CRISPR editing *in vivo*, they should provide at least proof of concept data that his strategy can work *in vivo* in their xenotransplantation model.

Our objective in this manuscript is to show that microglia are key pathogenic mediators of FRDA neuropathology. As stated by other reviewers, heterogeneity between individuals has been a challenge in the field, and having an isogenic cell line with the GAA repeat deletion is in our view a strength, correcting the FRDA phenotype in the process. We clarified this point in the revised manuscript. We fully agree that for a CRISPR editing strategy to move forward to full clinical translation, substantial additional work is needed. For instance, to transplant these corrected microglia into an FRDA murine model, which is a difficult model due to repeat expansion instability, would require crossing the FRDA mice with our multiply transgenic model (Rag2, Il2rg, CSF1, and Csf1r intronic deletion) and would be an extensive undertaking indeed. Our goal here is to utilize these lines in the key discovery of myeloid cells as *primary* mediators of significant FRDA pathology.

However, this work is also very relevant to the clinical translation of our gene editing approach as we are in the IND-enabling stage for CRISPR/Cas9-mediated gene editing hematopoietic stem and progenitor (HSPC) cell transplantation in FA. As HSPC-derived myeloid cells are known to differentiate into microglia-like cells in the CNS^{1,2}, it provides a mechanism of action for the success of gene edited HSPC transplant to prevent neurodegeneration in FA. (PMID: 35190726, PMID: 35197828)

2) The key experiment on this manuscript is the *in vivo* part, but it is very underdeveloped. Besides the CD68 quantification, are there any other indications that the several observations from the *in vitro* experiments are valid *in vivo*? Do microglia show alterations in phagocytosis, lipid peroxidation, etc? And do neurons display DNA damage or increased Caspase-3 expression?

We thank the reviewer for this question. We have now performed additional experiments and quantitation *in vivo* to address this point. First, we added 8-OHdG levels in our *co-culture* model as well as *in vivo* specifically in xMGs and in Purkinje neurons. 8-Hydroxydeoxyguanosine (8-OHdG) is a marker for oxidative stress, and it is used to detect oxidative damage of DNA as a consequence of ROS accumulation. We showed that not only *in vivo* FRDA xMGs retain their oxidative phenotype that we extensively characterized *in vitro* (fig. 2 and 4) but also that FRDA iMGs and xMGs trigger oxidative stress and DNA damage in healthy neurons (fig. 5d and g, fig. 6f and 7b). Additionally, elevated phagocytic marker expression CD68 is often used as a proxy of phagocytic microglia PMID: 768092, PMID: 23943781.

3) The entire manuscript is merely descriptive, and does not provide any mechanistic insights as to how 1) the FRDA mutation leads to microglial cell autonomous functional alterations, and most importantly 2) by which mechanisms the altered microglia can modify neuronal physiology. I feel this is a missed opportunity and more mechanistic experiments should be provided.

We do agree with the reviewer that in this manuscript we have not definitively untangled the mechanisms leading to the observed phenotypes both *in vitro* and *in vivo*. However, our new recent findings about the robust elevation of ROS and subsequent DNA damage in healthy neurons *in vitro* and *in vivo* helps in the understanding on putative mechanisms leading to healthy neuronal degeneration. We did edit the manuscript text emphasizing that due to the high sensitivity of neurons to oxidative stress, elevation of 8-OHdG levels might be a significant contributor to the degenerative phenotype here reported. Additionally, we have performed further conditioned media experiments (supplementary fig. 6) to query whether the neuronal stress/death phenotype is cell-cell mediated. Using multiple measurements including H2AX, caspase3, and neuronal blebbing, we find that the phenotype seen in coculture is not generated from conditioned media alone and is cell-cell interaction dependent. We feel this markedly brings forward the FRDA field to find that microgliopathy is a primary mediator of neuronal death through cell-cell microglial-neuronal interactions.

4) The *in vitro* part is generally robust and well performed. However, I noticed a number of aspects that need to be improved

a) In figure 1. It is not clear how many cells were quantified per human sample in figure 1a and b.

Sholl analysis was performed on 10 cells per patient from 3 patients per condition. The data were pooled, resulting in 30 cells per condition, or 60 cells total.

In panels f and g, there are clear differences in the density of cells based on the images provided. Cell density has a strong impact on iPSC microglia, both at morphological and phenotypic level. How do the authors rule out that there is no such impact in their system?

Cells were always plated at the same density during their differentiation phase as well as after replating onto slides when performing experiments. Therefore, the density of cells/volume is identical between different experiments. However, a significant challenge with microglia cells is their semi-adherent nature, which sometimes leads to loss on slides or other substrates during imaging/washing. This is the reason why multiple biological replicates of different independent differentiations and different cell lines were performed for each assay.

In this same line the quantification of CD68 and Iba1 in Figure 1g does not seem to be normalized per cell, therefore also introducing the confounding element of cell density. There are also discrepancies between the images and quantifications. From the images, the edited microglia seem to have as much (if not more) Iba1 signal than the FRDA cells; however the quantification does not show that. Why are the pictures from the other experimental not shown?

We apologize for any confusion and have clarified the methodology in the figure legends. The integrated density analysis was performed per cell and not per image, with a comparable number of cells analyzed between different conditions, for each biological replicate, at least 80 cells/line/replicate. The analysis is therefore not biased by cell number. We have replaced the overexposed representative image of Iba1 in 1g. Given space limitations we are showing FRDA

and gene edited images in the main figures for each experiment but have included them now as supplementary figure 2d. Given figure overcrowding additional images for carriers and controls are now included in the supplementary figures.

b) In figure 2. Again, the quantification of staining in vitro does not seem to be corrected by cell density, which may have a strong impact on the results. Why only pictures from only two experimental groups are displayed. The authors intent to claim that the Cas9 mediated excision of the pathogenic expansion normalized a series of readouts, but the representative examples of control cells are systematically omitted.

We apologize – the additional images of control and carrier were omitted due to the already rather large size of the figures. We have now included them as an additional supplementary figure. For all conditions, the analysis was performed *per cell* and it has been clarified in the figure legends to reflect this more clearly. For each biological replicate at least 80 cells were analyzed. As above, the images from control and carrier were omitted due to space but are now included in an expanded supplementary figure as requested. All conditions were always analyzed as indicated, *per cell*, and collapsed across technical replicates to only display independent biological replicates and each experiment was performed three times independently.

Figure 2g and i appeared to be quantified on the basis of ROIs as individual datapoints. This is considered pseudoreplication, and statistics should be performed at the level of independent replicates

We do apologize. We have now included more replicates for figure 3f and new analysis next to it. We performed additional electron microscopy specifically for figure 2, allowing for the improved analysis in 2g, however the quality of the EM was insufficient to accurately quantify the cristae structure in 2h. We have detailed the number of lines (two – three lines per genotype) included in the analysis within the figure legend.

c) In figure 3. In panel a it is shown that the mitophagy is actually not changed between the FRDA and the edited cells.

The FRDA microglia have reduced mitophagy (yellow) compared to the edited and control microglia (Fig. 3a). The yellow signal is the result of the colocalization between the mitophagy dye (red) and lysosome dye (green). The dye we used (MD01-10; Dojindo) will specifically accumulate in healthy mitochondria, producing a weak red fluorescence signal that will significantly increase at a pH4-5, therefore once the mitochondria (red) will fuse with the lysosomes (green dye) and will result in a yellow signal fluorescent signal.

The link between mitophagy and phagocytosis is unclear. These are two fundamentally unrelated processes, that could be impaired in an independent way due to energetic and metabolic deficits induced by the pathogenic mutation.

We apologize if we created any confusion. We are not stating that as a consequence of altered mitophagy, FRDA microglia are hyperphagocytic. Proper mitophagy in microglia also alleviates microglia inflammatory response and has an impact on microglia activation (PMID: 36034136). We have emphasized this in the text to make it clear that we are not trying to functionally link these two independent and robust findings.

The quantification in figure 3f suffers from pseudoreplication.

We have now removed the quantitation.

d) The co-culture experiments are interesting and very provocative. However, to completely prove the impact of microglia in FRDA, additional experiments with FRDA neurons cultured with non-mutant microglia should be performed.

We agree that FRDA neuronal crosstalk, and whether healthy microglia can rescue FRDA neurons is a key future question for the field that will require extensive future work. The discovery here is focused on the pathogenic contribution of FRDA microglia to neurodegeneration, which we have undertaken with multiple models, coculture, conditioned media, and xenotransplantation. Whether healthy microglia can rescue FRDA neurons is outside the scope of this study. To extend on our mechanistic data, we have now included 8OHdG staining in our co-culture model to measure DNA damage as a result of oxidative stress to further characterize the neurodegenerative phenotype caused by FRDA microglia. Additionally, we have completed further characterization of the neuronal phenotype in the presence of microglia conditioned media which includes data looking at caspase-3, neuronal morphology and H2AX as a proxy for DNA damage, finding that cell-cell interactions are necessary for neuronal injury.

e) Across the different readouts shown here, the carrier line only partially mimics the FRDA group. What are the implications of this observation for the human disease?

Carriers harbor the mutation in a single allele which is not sufficient to cause a phenotype. They are considered healthy individuals. However, whether these subtle microglial deficits are sufficient to cause premature microglial aging or contribute to late onset cognitive or other deficits remains to be explored. We were also surprised to find that carriers did not have entirely normal healthy microglial phenotypes given that carriers are presumed 'healthy' but have reported our data for full clarity. We have extended on this in our discussion.

5) The in vivo experiments are very interesting, but also very limited. It could greatly benefit from a larger sample size and more readouts. There are also some discrepancies that need to be addressed. For example, in figure 6 there seems to be inconsistencies between the images shown in panels b and f.

We have now added untransplanted mice into our cohort and analysis as an additional control. Each point in figures 6b and 6f represent a different xenotransplanted mouse as an independent biological replicate, with statistical significance identified. Therefore, each experiment includes 3-4 mice/genotype. We have also replaced the representative images in 6b as requested, and have split Figure 7 to further quantitate the neuronal phenotype using orthogonal Purkinje cell markers as well as 8OHdG as a marker of oxidative stress in Purkinje cells as a result of FRDA microglia.

Whereas in f it is shown PC loss, this is not clear in the images in panel b. Which one of the two observations is correct? The PC loss should be also be confirmed with additional stainings to rule out the findings are confounded by a downregulation of calbindin.

We apologize for any confusion and better representative images have been selected— We assessed neuronal loss by specifically quantitating the number of Calbindin⁺ neurons in 100µm of PCL therefore concluding that lower Calbindin⁺ neurons reflect neuronal loss. Additionally, we

also stained Purkinje cells using IP3R and we again confirmed a similar reduction of IP3R+ neurons in the Purkinje cell layer in FRDA xenotransplanted mice compared to healthy and untransplanted control. We also quantified the number of granule cell neurons using NeuN and we showed that neuronal death is not uniquely occurring in the Purkinje cell layer but rather a broad cerebellar neuronal degeneration might occur as a consequence of FRDA xMGs. These data are displayed in fig. 7a, c and Suppl. Fig 7c.

REVIEWER COMMENTS

Reviewer #1 (Remarks to the Author):

The authors have addressed the majority of the concerns raised in the previous round of review, particularly with regard to the in vitro experiments, which are now much more complete and convincing. While I still have some reservations about the in vivo data, I acknowledge the additional efforts made and believe the current version of the manuscript represents a significant improvement. I have no further major concerns.

We thank the reviewer for their kind comments and support.

Reviewer #2 (Remarks to the Author):

Reviewer #3 (Remarks to the Author):

The revised manuscript has substantially improved the quality of the presented data and supports the claimed conclusions.

We thank the reviewer for their kind comments and support.

Reviewer #4 (Remarks to the Author):

Reviewer #5 (Remarks to the Author):

Thanks to the authors for the revised version of their manuscript. The authors have addressed most of my comments regarding the in vitro part of the paper. However, there are still some critical points that need additional data. In the current form, I do not think the main message of this manuscript – microglia can be primary inductors of neuronal cell death in this form of Ataxia – is well supported. Below a list of aspects that can be improved.

1) Although I understand the limitations, the current manuscript still does not provide (or even speculate) on what potential mechanisms are involved in this putative physical induction of neuronal death via microglia. There are previous reports showing that during brain development, microglia can promote the degeneration of Purkinje cells in the absence of any, and mediated by physical interactions mutations (Marin-Teva et al. 2004). I suggest the authors do a thorough search in the literature.

We agree with the reviewer that the current manuscript does not fully dissect the precise mechanisms underlying microglia-mediated neuronal death, as this lies beyond the scope of our study and will require further and deeper investigations. We reviewed the suggested reference which hypothesizes that *the presence of microglia is needed for the elimination of immature and caspase-3⁺ Purkinje Cells (PCs) during cerebellar development*. This reference claims that microglia are required to phagocytose Caspase-3 positive PCs, but their data do not clarify whether the initiation of PC death per se is cell autonomous, due to extracellular signals from other cerebellar neurons or glia such as astrocytes, or initiated by microglia. Indeed, as the authors emphasize, the slicing process used to generate slice

cultures interrupts the initial olivocerebellar inputs to PCs and severs the axons projecting to PC targets in the deep cerebellar nuclei, a perturbation that can potentially induce PC death. Indeed, treatment with insulin growth factor-1 (expressed by olivary neurons and deep cerebellar nuclei during development) in cultured slices improves PC survival within the same experimental scenario (PMID: 1658250 and 11978830), suggesting that the initiation of PC death could indeed be partially initiated as a technical complication of the experimental perturbation.

However, since the authors could not exclude a microglia-mediated initiation leading to PC death and per the reviewer's request, we examined whether aberrant microglia activity during cerebellar development could underlie PC loss in our experiments. To this end, we quantified Purkinje cell numbers in juvenile mice (P21), immediately after the major developmental phase of the cerebellum is concluded. As shown in the new Supplementary Figure 7d, we found a comparable number of PCs at this stage, finding that PC loss does not occur during development and could indeed be caused by a more nuanced neurodegenerative mechanism. Consistent with the reviewer's noted reference and PMID: 33450391, there was a tendency towards more Purkinje cells in untransplanted animals compared to FRDA mice, with no difference among the three xenotransplanted groups. This, together with our *in vitro* and *in vivo* findings—demonstrating elevated ROS production and consequent DNA damage in healthy neurons exposed to FRDA microglia—supports our hypothesis that neuronal degeneration could be ROS-mediated and occurs during adulthood, rather than during development. We have updated and included additional references in the revised manuscript as requested.

2) The proposed main message of this manuscript is not well supported by the data. I think that the authors have provided a fantastic characterization of the microglia, and with no doubt one can claim that FRDA mutations primarily affect microglia. However, the claims of microglial mediated neuronal toxicity are not well supported. If the authors want to still claim that, they should provide additional data. Below I suggest a list of potential experiments.

a. Co-culture experiments using mutant iPSC-neurons and wild type microglia. This would address whether the impact of mutant microglia is relevant when neurons also carry a mutation. It could well be that the neuronal autonomous effect of the disease causing mutation will result in neurodegeneration regardless of the contribution of dysfunctional microglia.

The question of whether and how diseased neurons may be therapeutically recovered by healthy microglia is an entirely separate question with separate mechanistic implications that lie far outside the scope of this manuscript. We would postulate that healthy microglia may very well ameliorate aspects of FRDA neuronal dysfunction that may occur through diverse potential mechanisms of protein or mitochondrial transfer that have yet to be addressed in the field and are outside the scope of this manuscript. We do not make claims that microglia are the *only* mediator of neurodegeneration in FRDA, but rather the novel idea that they may be a significant independent mediator. The novel question we have sought to shed light on in this manuscript is whether FRDA microglia are independent contributors to neurodegeneration in and of themselves, and have done so through iPSC microglia, complex cocultures using two different differentiation methods (through neural progenitors and through directly induced neurons) and through xenotransplantation for a robust dataset. We have carefully edited the discussion to ensure that we do not make claims outside the scope of our data, as our own published data has shown FRDA neuronal phenotype (PMID 38420191), and we do not make claims of exclusivity.

b. Is the cell death induction specific to Purkinje cells. I suggest performing a brain wide experiment looking at different anatomical regions to determine if this is a true selective vulnerability. I would presume that, if microglia are so dysfunctional, there would be other brain alterations.

We thank the reviewer for this comment. In our study, we demonstrate that neuronal loss is not limited to cerebellar Purkinje cells but also extends to granule cell death as shown in Figure 7c. We agree that there is growing evidence suggesting that Friedreich's ataxia exerts neurodegenerative effects beyond the cerebellum. Nonetheless, because of the nature of the disease, we chose to focus on the cerebellum to compare our findings with the substantial recent evidence highlighting cerebellar microglia alterations and

their impact on local neuronal populations (PMID: 35401081, 38631900, 39111701, 29257745, 39100202). The involvement of other areas such as the frontal lobe, thalamus, and hippocampus remains incompletely characterized in patients, making it difficult to establish direct human-specific correlates for murine phenotypes and therefore lie outside the scope of this study.

c. The *in vivo* experiment is very shallow and it is the key argument that supports the main claim of this manuscript. I have serious concerns that the observations are not the result of confounding factors. The authors used FIRE-KO mice for their experiments, which are a model of CSF1R-related leukodystrophy (REF). ALSP typically presents with cerebellar ataxia. Alterations in cerebellar development and PC have been also observed in several mouse models (Kana et al. 2019; Marin-Teva et al, 2004; Nakayama et al. 2019, etc.). There is a substantial risk that the mouse model itself presents with cerebellar alterations, that can then be corrected with wild type but not FRDA microglia. Although this could be an interesting finding, it is not what the authors propose here. I suggest the authors check the cerebellar phenotype of their host mice in depth, and perform additional confirmatory experiments in mice that do not present a developmental lack of microglia. In fact, the only image provided in S7c shows aberrant distribution, uneven size and morphological altered PCs in untransplanted vs. control mice.

The FIRE murine model carries an intronic enhancer deletion in the murine *Csf1r* gene that leads to genetic depletion of murine microglia. It is not a patient disease-specific mutation, and there is disagreement as to what degree it represents CSF1R-related leukodystrophy (PMID 40638739). The FIRE mouse does develop age-related features of leukodystrophy, however FIRE mice do not have pathology until 8+ months of age (PMID 38897208 and 38897209). The data included herein is at 2 months of age. In keeping with the published data and *including mice that are not xenotransplanted in all experiments*, we show that xenotransplanted FRDA microglia led to Purkinje cell and granule cell loss when compared to untransplanted FIRE mice, control transplanted FIRE mice and gene-corrected transplanted FIRE mice (Fig. 7). There is currently no published mouse model in which microglia can be thoroughly and completely depleted postnatally allowing for microglial xenotransplantation. Our own and other published work finds that xenotransplantation into a model with remaining murine microglia leads primarily to microglia in the hippocampus and thalamus (e.g. PMID 31375314, among others), without engraftment in the brainstem or cerebellum therefore would not be amenable to addressing this question. We agree that in untransplanted mice with no microglia, Purkinje cell distribution and morphology is altered consistent with the extensively published role of microglia in cerebellar maturation. We appreciate the reviewer's reference to the known contribution of microglia to Purkinje cell morphology and distribution (PMID: 31350310), a phenotype that is appreciated in our representative images of untransplanted mice. Given this, we did not investigate these PC morphological phenotypes in this study.

To address questions about whether our murine model has cerebellar alterations that can be corrected by wildtype but not FRDA microglia, we have performed additional experiments in all four groups (untransplanted, and xenotransplanted with FRDA, edited, and control microglia) performing histology in mice at postnatal day 21, at the end of the cerebellar developmental period (new Supplementary Figure 7d). Here we show that there is a tendency toward more PCs in the untransplanted group, consistent with the known contribution of microglia in removal of Purkinje cells during development, with no statistical difference between the three xenotransplanted groups. These data suggest that the neuronal loss we report in adult animals is genotype-specific and post-developmental consistent with the main findings of the manuscript. In addition, several other publications have reported comparable numbers of PCs in wildtype control mice to our study, and a similar Purkinje cell number is also found in a rat model with *Csf1r* KO suggesting that PC numbers are intact and that the neurodegenerative phenotype we are reporting here is a direct consequence of FRDA microglia (PMID: 21870131, 26748090, 33450391).

3) "Notably, CRISPR/Cas9-mediated correction of the GAA repeat reverses microglial defects and mitigates neurodegeneration." This sentence in the abstract is misleading. The authors do not use any CRISPR-based therapeutic approach, but simply correct and transplant non-mutant lines. Please adjust the abstract to fit to the data provided.

We respectfully disagree with the reviewer. The CRISPR/Cas9 mediated GAA repeat correction utilized herein has been published by our group (PMID 32462051) in hematopoietic stem cells and is currently in the IND-enabling study stage to start a clinical trial for autologous gene edited hematopoietic stem cell transplant for Friedrich's ataxia. Multiple patient cell lines utilized in this manuscript were CRISPR/Cas9 edited to remove the GAA repeat using the same guides as published for hematopoietic stem cells and utilized in all experiments throughout including in xenotransplantation. All analysis in vivo includes four groups – unengrafted, healthy control iHPCs-engrafted, FRDA disease iHPCs-engrafted, and CRISPR/Cas9 gene edited iHPCs-engrafted.

4) In figure 6b, the authors show an increase density of microglia in the areas where there is degeneration. Could it be simply that the dysregulated density of microglia is altering the physiology of the tissue?

This would be speculation since it is not possible to phenotypically isolate microglial density from other aspects of FRDA microgliopathy. We would speculate that the observed phenotype is not only due to an increase in microglial numbers, given the other presented data including microglial morphological differences (Fig. 6c) indicative of a more amoeboid phenotype, increase in antigen presentation (Fig. 6e, S7b) and CD68 (Fig. 6d), and increase in microglial 8OHdG+ (Fig. 6f) indicative of increased oxidative DNA damage that altogether suggest there are not simply more microglia but that FRDA microglia exhibit microgliopathy. Additionally, we find a reduction in NeuN+ neurons in the granule cell layer of mice transplanted with FRDA microglia compared to all other groups. Of note, microglia density in the granule cell layer was comparable between different cell lines (Fig.6b) supporting the hypothesis that the hyperinflammatory phenotype of xMGs leads to neuronal degeneration rather than microglial density or differential distribution across cerebellar layers.

5) There are still quantifications that suffer from pseudoreplication, such as 2h. Please correct this.

As indicated in our prior response to reviewers, we performed additional electron microscopy to address pseudoreplication. Unfortunately, our additional electron microscopy did not have the same superb resolution as our first experiments, therefore we could not confidently quantitate cristae ultrastructure in our second set of samples - but did quantitate mitochondrial size, for example for which the quality was sufficient. We have removed this quantitation in the revised manuscript.

Reviewer #5 (Remarks to the Author):

The authors have addressed my points satisfactorily. Thanks for the extra effort

We thank the reviewer.